# Assessment of air pollution and air quality perception mismatch using mobility-based real-time exposure

**Wanying Song[1], Mei-Po Kwan** [ORCID]**[1,2,3]\*, Jianwei Huang[1]**

**1** Institute of Space and Earth Information Science, The Chinese University of Hong Kong, Shatin, Hong Kong, China, **2** Department of Geography and Resource Management, The Chinese University of Hong Kong, Shatin, Hong Kong, China, **3** Institute of Future Cities, The Chinese University of Hong Kong, Shatin, Hong Kong, China

\* mpk654@gmail.com

**Data Availability Statement:** Data cannot be shared publicly because they contain potentially identifying information that cannot be fully de-identified. Note that the dataset has participants' GPS tracks and activity locations (including home

## Abstract

Air pollution poses a threat to human health. Public perceptions of air pollution are important for individual self-protection and policy-making. Given the uncertainty faced by residence-based exposure (RB) measurements, this study measures individuals' real-time mobility-based (MB) exposures and perceptions of air pollution by considering people's daily movement. It explores how contextual uncertainties may influence the disparities in perceived air quality by taking into account RB and MB environmental factors. In addition, we explore factors that are related to the mismatch between people's perceived air quality and actual air pollution exposure. Using K-means clustering to divide the $PM_{2.5}$ values into two groups, a mismatch happens when the perceived air quality is poor but the air pollution level is lower than $15.536 \mu g/m^3$ and when the perceived air quality is good but the air pollution level is higher than $15.608 \mu g/m^3$. The results show that there is a mismatch between air pollution exposure and perception of air pollution. People with low income are exposed to higher air pollution. Unemployed people and people with more serious mental health symptoms (e.g., depression) have a higher chance of accurately assessing air pollution (e.g., perceiving air quality as poor when air pollution levels are high). Older people and those with a higher MB open space density tend to underestimate air pollution. Students tend to perceive air quality as good. People who are surrounded by higher MB transportation land-use density and green space density tend to perceive air quality as poor. The results can help policymakers to increase public awareness of high air pollution areas, and consider the health effects of landscapes during planning.

## 1 Introduction

Air pollution exposure is one of the most important risk factors for human health. Past studies have found that exposure to fine particulate matter was associated with overweight and obesity, cardiovascular disease, early death, and neuro-developmental outcomes [1, 2]. Exposure

and workplace locations), which can be used to discover the identity of individual participants using a process called spatial reverse engineering (e.g., by linking high-resolution geospatial data with other data such as census or survey data to discover the identity of specific individuals). The Survey and Behavioral Research Ethics (SBRE) Committee of the Chinese University of Hong Kong has imposed this restriction. Please contact the SBRE at fssc02@cuhk.edu.hk for inquiries about data availability or access.

**Funding:** Mei-Po Kwan was supported by grants from the University Grants Committee of the Hong Kong Research Grants Council (https://www.ugc.edu.hk/eng/rgc/) (General Research Fund Grant no. 14605920, 14611621, 14606922; Collaborative Research Fund Grant no. C4023-20GF; Research Matching Grants RMG 8601219, 8601242), and a grant from the Research Committee on Research Sustainability of Major Research Grants Council Funding Schemes (3133235) of the Chinese University of Hong Kong (https://www.cuhk.edu.hk/english/index.html). The funders had no role in study design, data collection and analysis, decision to publish, or preparation of the manuscript.

**Competing interests:** The authors have declared that no competing interests exist.

to air pollution also influences mental health, including cognitive function, sleep disturbance, subjective well-being, aggressive behavior, social stress and depression [3, 4]. In recent years, people are thus paying more attention to air pollution and making some efforts to mitigate its harmful health impacts, such as wearing masks using air purifiers and installing exhaust hoods in kitchens. However, some people may not be able to respond to severe air pollution quickly due to a lack of relevant knowledge of air pollution. They may not understand or recognize the harmful impacts of air pollutants. Therefore, exploring whether people's perceptions of air pollution align with the actual (i.e., objective) levels of air pollution is crucial for responding to its impacts on people's physical and mental health.

Research on environmental risk perception seeks to understand the difference between expert and lay perceptions of risk [5]. Experts measure and learn about air pollution based on its concentrations, components, and statistical evidence of air pollution's adverse health impacts on human beings. While lay public perceptions of air pollution are affected by people's sensory clues, their socioeconomic status, health status and habits, and so on [6]. Specifically, the visibility and odors of air pollutants are found to be the fundamental elements of public perception [7]. For example, previous studies showed that there was a correlation between air dust perception and air quality perception, indicating that people perceive air quality by the visual effects of particles and dust [8–10]. People can recognize air pollution sources such as garbage, smoke and low visibility of the air [6]. They can also sense air pollution through the pungent smells from drainage channels, factories and toilets in communities [11]. Therefore, sensory awareness plays a fundamental role in forming public the perception and subsequent responses to air pollutants [12].

Studies have also found that air pollution perception is influenced by socioeconomic and demographic factors. For example, older, more conservative, female, and ethnic-minority people who have been exposed to environmental risks are likely to perceive that they face greater environmental risks [13]. Perceptions of poor air quality were associated with low-income, married and divorced/separated/widowed people [14]. Well-educated people were likely to be more informed about the adverse health effects of air pollution, which may lead to higher levels of perceived risk [15, 16]. Watching local TV news increased people's knowledge about air pollution and further increased perceived environmental risks [13].

Besides socioeconomic factors, people's health status and previous experiences with air pollution may also influence how they assess air quality. Those who have suffered from severe haze pollution may be more likely to discern air pollutants and take measures to protect themselves during heavily polluted days [15]. Self-reported health status is also linked to the perception of air pollution since people with illness would pay more attention to the harmful sources that might threaten their health [17]. People who had a medical history of hay fever and symptoms of headache, dizziness, breathing difficulties, or exhaustion perceived the air to be dustier than other groups [17, 18].

Air quality perception is also influenced by environmental conditions. Higher humidity might lead to lower visibility, which may lead to people's perceptions of poor air quality given their visual sense of air quality [19]. People perceived air as very clean when the weather is cool and dry, while people perceived warm and highly humid air as stuffy even if the air is clean [20]. Green space is found to mitigate people's negative perceptions of air quality [21]. People are also aware that poor air quality had negative health consequences in commuting [22]. Some people who suffered from previous air pollutants would change their transportation modes to reduce harm, meaning that they are aware of transportation as an air pollution emission source [15].

People tend to believe that the air quality in their neighborhoods is better than that in other neighborhoods, known as the "neighborhoods halo effect" [5]. Besides, it is found that people's

perceptions of air pollution relate to which activity they conduct. People tend to perceive air pollution to be less serious when the activities' benefits outweigh the harms caused, such as running outdoor [5]. Although previous studies have provided a fundamental understanding of people's perceived air pollution, they did not explore whether people would overestimate or underestimate air pollution in their surroundings and which groups of people would overestimate or underestimate air pollution. Understanding whether people's perceptions of air pollution align with actual air pollution levels would provide crucial knowledge of public attitude towards air quality. Therefore, this study aims to explore the influencing factors of people's perception of air pollution, whether there is a mismatch between actual and perceived air pollution, which social groups tend to misinterpret air pollution, and the environmental factors that contribute to the mismatch.

One important factor deserves special attention when exploring the association/mismatch between actual and perceived levels of air pollution: the geographic context used to delineate mobility-dependent environmental factors. Past studies usually adopted static methods to measure air pollution from monitoring stations and measured environmental factors, such as green space, traffic conditions, or facilities around people's residential neighborhoods [23, 24]. However, using people's residential location as the geographic context may not reveal how much air pollution they are actually exposed to because people travel to various locations and are also exposed to these non-residential contexts [25, 26]. Using only the residential location thus cannot capture how people interact with and interpret the air quality in different geographic or activity contexts [9]. People's awareness of good or poor air quality may be influenced by environmental factors in the places they visited and at locations other than their residential neighborhoods. Air pollution concentrations, traffic conditions, building density, green space, and social facilities vary over space and time [27]. Therefore, air pollution levels and environmental factors could be significantly different when measured using a static versus dynamic approach, which further leads to the uncertain assessment of the links between air pollution and air quality perception. Previous studies' ignorance of mobility-based (MB) geographic contexts would lead to the uncertain geographic context problem (UGCoP) [28]. The UGCoP is the problem that the measurement of environmental exposures might be different due to the use of different delineations of geographic or environmental context, which could further affect the conclusions of environmental impacts on health outcomes.

There are several methods for tackling the potential bias caused by static air pollution measurements. The first and direct measurement method is measuring individual real time air pollution exposure using portable sensors. For example, Zhang et al. used the RB and MB approaches to estimate environmental exposures and explored the association between environmental exposures and mental health [29]. They found that the MB approach has a higher explanatory power for mental health than the RB approach. Another method for estimating individual MB exposure is using exposure equation based on people's body weight, respiratory rates, indoor to outdoor air pollution concentrations, and time-activity patterns based on air pollution monitoring stations data [15]. Studies usually obtained hourly or daily air pollution monitoring station data and estimated air pollution in the whole study area based on spatial interpolation, combined with satellite data, or air dispersion models [27, 30, 31]. A study estimating individual air pollution exposure by monitoring stations air pollution data and exposure equation found a correlation between individual exposure levels and risk perception of $PM_{2.5}$ [15].

To sum up, this study seeks to explore: (1) the social, economic, demographic and environmental factors that influence people's perceptions of air pollution, and (2) whether there is a mismatch between actual (objectively measured) air pollution and perceptions of air pollution and how addressing the UGCoP could mitigate such mismatch. This study measured MB air

pollution to reduce the potential bias caused by using only residence-based (RB) exposure in exploring the influencing factors of air pollution perception and the mismatch between objective and perceived air quality. Specifically, we used GPS and portable sensors in our study to collect real-time data on participants' location and exposure to particulate matter, temperature, and relative humidity (RH). In other words, participants' exposures when conducting their daily activities in specific environments are recorded, reflecting the most accurate measurement of real-time air pollution exposure. In this way, the association between actual and perceived air quality can be investigated in depth with high precision. This study also adopted two methods to measure environmental factors, residence-based (RB) and mobility-based (MB) methods to explore how UGCoP could affect the association between influencing factors of air pollution perceptions and the mismatch between actual and perceived air quality.

## 2 Dataset and methods

### 2.1 Study area

The study area for this research is Hong Kong (HK). 40% of the land in HK is designated as country parks or special areas for nature conservation [32]. Only around 25.40% of the land resources are built-up areas [33]. Therefore, HK has a high building density and crowded urban areas that are surrounded by open, natural green landscapes and wildlife habitats. It has few integrated landscapes in the urban areas [34, 35], such as the districts of Sham Shui Po, Kwun Tong, Kowloon City and Yau Tsim Mong [36]. Due to the high population and building density and extensive use of the transit-oriented development (TOD) model [37], approximately 90% of the passenger trips in HK are made by public transport, which is among the highest in the world [38, 39]. Private cars carry only 16% of the total daily road-based passenger boarding but account for about 40%-70% of the total traffic flow on most of the major roads, which causes serious traffic congestion [40]. The average traffic speed on Hong Kong Island (HKI) is the lowest among the three regions (HKI, Kowloon (KLN) and the New Territories (NT)) [40]. In HK, the primary sources of air pollutants are marine vessels, power plants, and motor vehicles [41]. The two greatest air pollution challenges are local street-level pollution and regional smog.

Among all the communities in HK, we chose Sham Shui Po (SSP) and Tin Shui Wai (TSW) as the study communities (Fig 1). SSP is one of the oldest developed areas in HK, with a very high population and building density (e.g., an average population density of 62,695 per km$^2$ in 2016 [42]). SSP also has the lowest median household income in the city [43]. SSP has many sub-divided units (multiple subdivisions within one apartment), increasing the vulnerability of residents to infectious diseases because of the inevitable close contact and lack of adequate space for physical distancing [44]. SSP has lively economic activities, such as street markets, electronic outlets, fabric stores, restaurants and food vendors [45]. TSW, being one of the third-generation new towns, was developed as a residential town to provide private and public housing to mitigate the high population densities in old urban areas [45, 46]. The northern part of TSW was mainly developed for public housing [46]. TSW has a population density of 77,162 per km$^2$ [47]. TSW has the sixth lowest median household income among the 18 districts in HK [43]. Since the northern part of TSW is near ecologically sensitive areas, lower-density housing developments are located in the north and east, whilst higher-density housing developments are concentrated in the south and west in the northern part of TSW [46]. Due to the failure to be self-contained and a spatial mismatch between housing and job opportunities, people in TSW often travel long distances for work [47, 48].

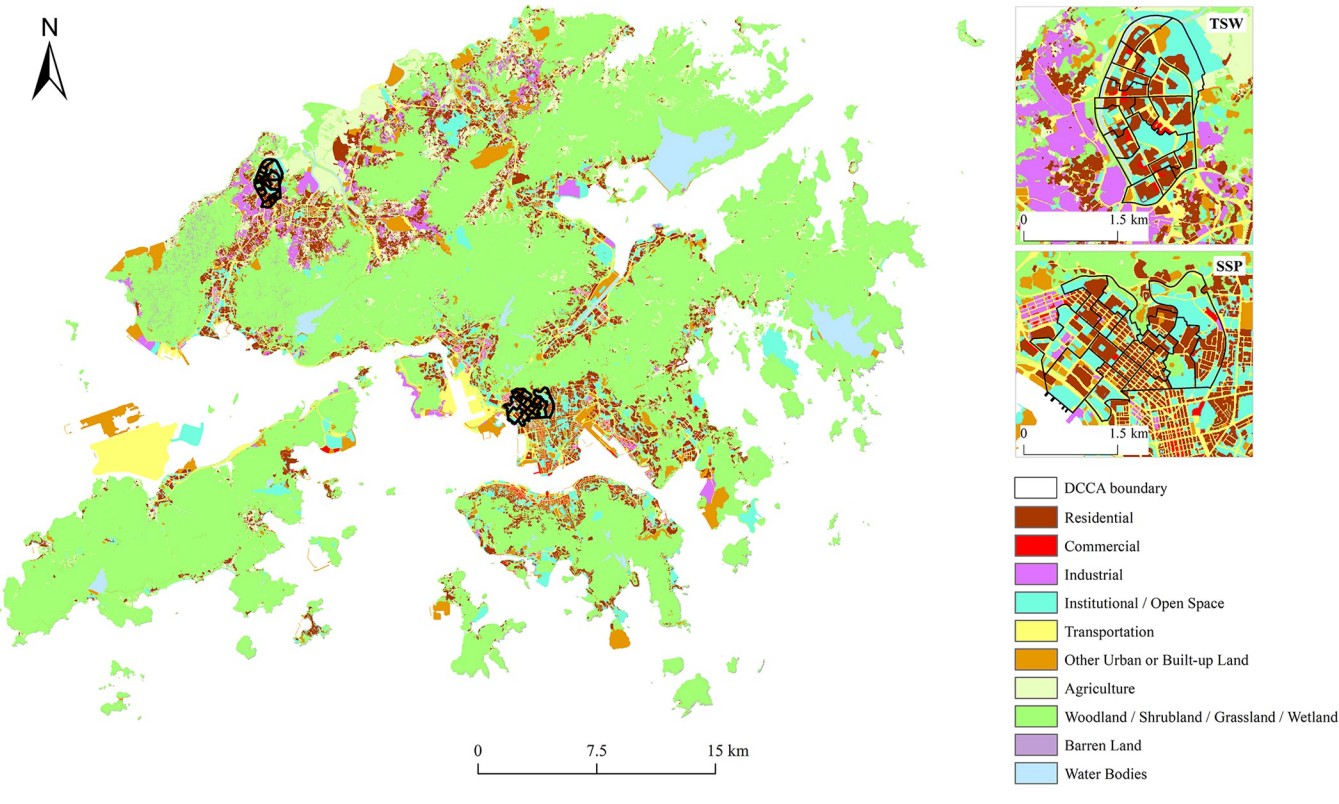

**Fig 1. Land use of the study area (Hong Kong).**

## 2.2 Data collection

We collected the data for this research from March 2021 to September 2021 through a stratified quota sampling survey. In each of the two communities (SSP and TSW), participants were recruited based on their age, gender, income and employment status from census sociodemographic data to ensure the representativeness of the samples. A total of 221 participants were recruited and 210 participants with adequate and valid sensor records were included in this study. In the survey, each participant carried an air pollutant sensor and a smartphone equipped with GPS all the time for two consecutive days. The two days include one weekday and one weekend day (i.e., Friday and Saturday, or Sunday and Monday) so that real-time air pollution measurements in the two days can represent people's exposure in a usual week. The sensors record $PM_{2.5}$ concentrations, relative humidity, and temperature at 1-second intervals. Real-time location information at 1-second intervals was also recorded by the GPS. Besides, participants were asked to fill out a questionnaire, which collected data about their socioeconomic status, perceptions of air pollution, and self-reported physical and mental health. The study protocol and survey instruments were reviewed and approved by the Survey and Behavioural Research Ethics Committee of the authors' university, and informed consent was obtained from all participants. During our survey, which was conducted in low-risk periods of COVID-19 in Hong Kong, there were no significant mobility restrictions compared to previous waves. When compared to the baseline day (the median value for the five-week period of January 3 to February 6, 2020), COVID-19 community mobility reports showed a 36.5% increase in visits to grocery and pharmacy stores, a 6.4% increase in visits to residential areas, a 9.3% decrease in visits to retail and recreation venues, a 7.3% decrease in visits to transit

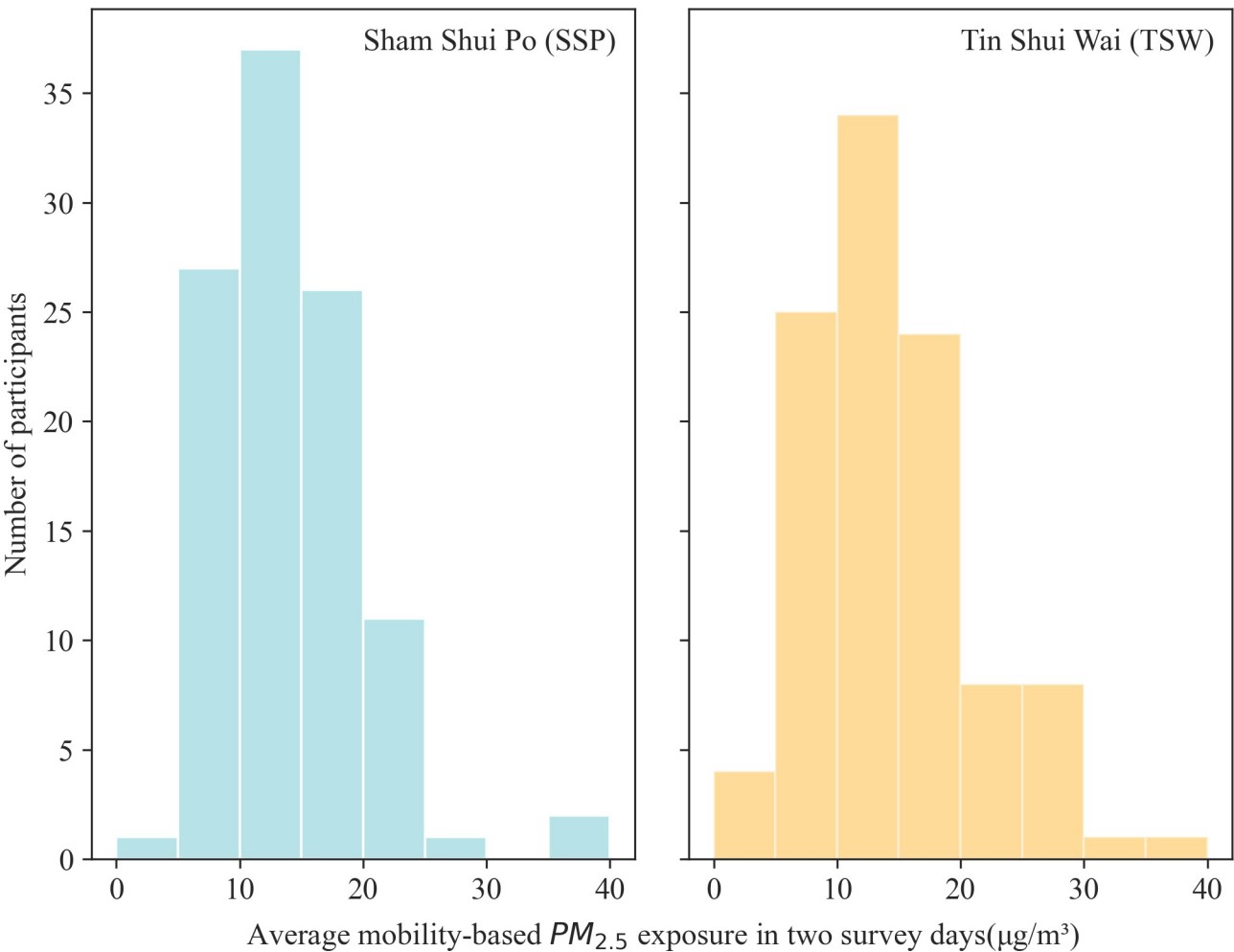

**Fig 2. Average PM$_{2.5}$ exposure in the two survey days in SSP and TSW.**

stations, an 18.1% decrease in visits to parks, and a 5.8% decrease in visits to workplaces on average [49]. In 2020, industrial activities decreased, air pollution concentrations decreased, and air quality improved due to COVID-19 [50]. However, as the epidemic stabilized in 2021, domestic economic activities and electricity demand rebounded, resulting in similar average annual levels of PM$_{2.5}$ (recorded by stationary monitoring stations) and increased levels of PM$_{2.5}$ (recorded by roadside stations) compared to 2020 [50, 51].

**2.2.1 Mobility-based air pollution exposure measurement.** We used the portable sensor AIRBEAM-2 (https://www.habitatmap.org/airbeam) to record each participant's real-time air pollution exposure due to its relatively good performance [52, 53]. Since portable sensors may sacrifice accuracy for convenience, we calibrated the portable sensors using DustTrak as a reference sensor based on a model we previously established [54]. The PM$_{2.5}$ concentrations were aggregated into 1-minute intervals based on the mean value to reduce minor fluctuations, and each participant's exposure to PM$_{2.5}$ is measured by the average mean value of exposure for the two survey days [55]. The histograms of the average mobile exposure of the two communities in the two survey days are shown in Fig 2. They indicate that the average exposure is right-skewed, with more values concentrated on the left side of the distribution where the maximum value falls into the range of 35–40 μg/m$^3$. Most of the participants are exposed to 5–20 μg/m$^3$

**Table 1. Participants' perception of air pollution in the study communities.**

| Neighborhood | Not at all | Not serious | Not very serious | A little serious | Serious | Very serious |
|---|---|---|---|---|---|---|
| Code | 1 | 2 | 3 | 4 | 5 | 6 |
| SSP | 0.0% | 3.8% | 17.6% | 19.5% | 9.0% | 0.0% |
| TSW | 0.0% | 7.1% | 19.5% | 17.6% | 5.7% | 0.0% |

of $PM_{2.5}$. There are more participants in SSP than in TSW with air pollution exposures between 5–25μg/m³ while more participants in TSW are in the value range from 25–35μg/m³. In the following sections, we use $PM_{2.5}$, air pollution exposure, or air quality interchangeably, and they have the same meaning.

**2.2.2 Measurement of air pollution perceptions.** In the survey, participants' perceptions of air pollution were assessed by asking "How serious is air pollution in the places you visit in a usual week?". The answer was recorded on a six-point scale, with 1 representing "not at all" and 6 "very serious." The basic statistics of participants' air pollution perceptions shown in Table 1 indicate that a higher percentage of people in SSP think air pollution is "a little serious" and "serious" while more people in TSW think air pollution is "not serious" and "not very serious".

**2.2.3 Measurement of mental health status.** In the questionnaire, participants reported their mental health symptoms through four response items, including feeling "nervous", "worried", "depressed" and "no interest in things". How frequently participants experienced such feelings over the past two weeks was recorded on a six-point scale where 1 represents "never have such a feeling" and 6 represents "always have such a feeling". The mental health score was calculated by summing all the corresponding items' scores [56]. The Cronbach's alpha of the mental health score is 0.887, indicating good internal consistency and validity. This is also reflected in Fig 3, which indicates that the distributions of all mental health responses have similar patterns, where the highest proportion of all mental health response items is "sometimes".

We summarized the average and median values of each mental health response item. The average values of the response items range from 3.167 to 3.705, fluctuating around 3.5((1+6)/2). The median value of "nervous" and "no interest" is 4, indicating that more than 50% of participants reported they "often" or "always" have such a feeling. The median value of "worried" and "depressed" is 3, indicating that more than 50% of participants seldom feel worried or depressed.

**2.2.4 Measurement of residence-based and mobility-based environmental factors.** Environmental factors in this study are specific types of land use or facilities: transportation land-use density, green space density, open space density, facilities and population density. Data were obtained via the HK Government's GeoData Portal [57]. Transportation land-use density, green space density, and open space density were calculated based on the 10m×10m raster land utilization data in HK (2020), which have 27 land use types. Green space was represented by "grassland", "shrubland", and "woodland" out of 27 land-use types. Open space is represented by "open space and recreation" out of 27 land-use types. Open space includes many elements, such as parks, playgrounds, sports centers, pools, conservation areas, picnic/barbeque spots, beaches, and so on [58]. Transportation land-use is represented by four land-use types, namely "roads and transport facilities", "railways", "airport", and "port facilities". The facility data were obtained from the Nearby API in the HK GeoData Store. This API can identify facilities within 1 km of a given location, including car parks, hostels, toilets, post boxes, convenience stores, shopping malls, markets, and so on. We covered the spatial extent of HK with 1415m×1415m grids (note that the diagonal of a grid with a 1km width is

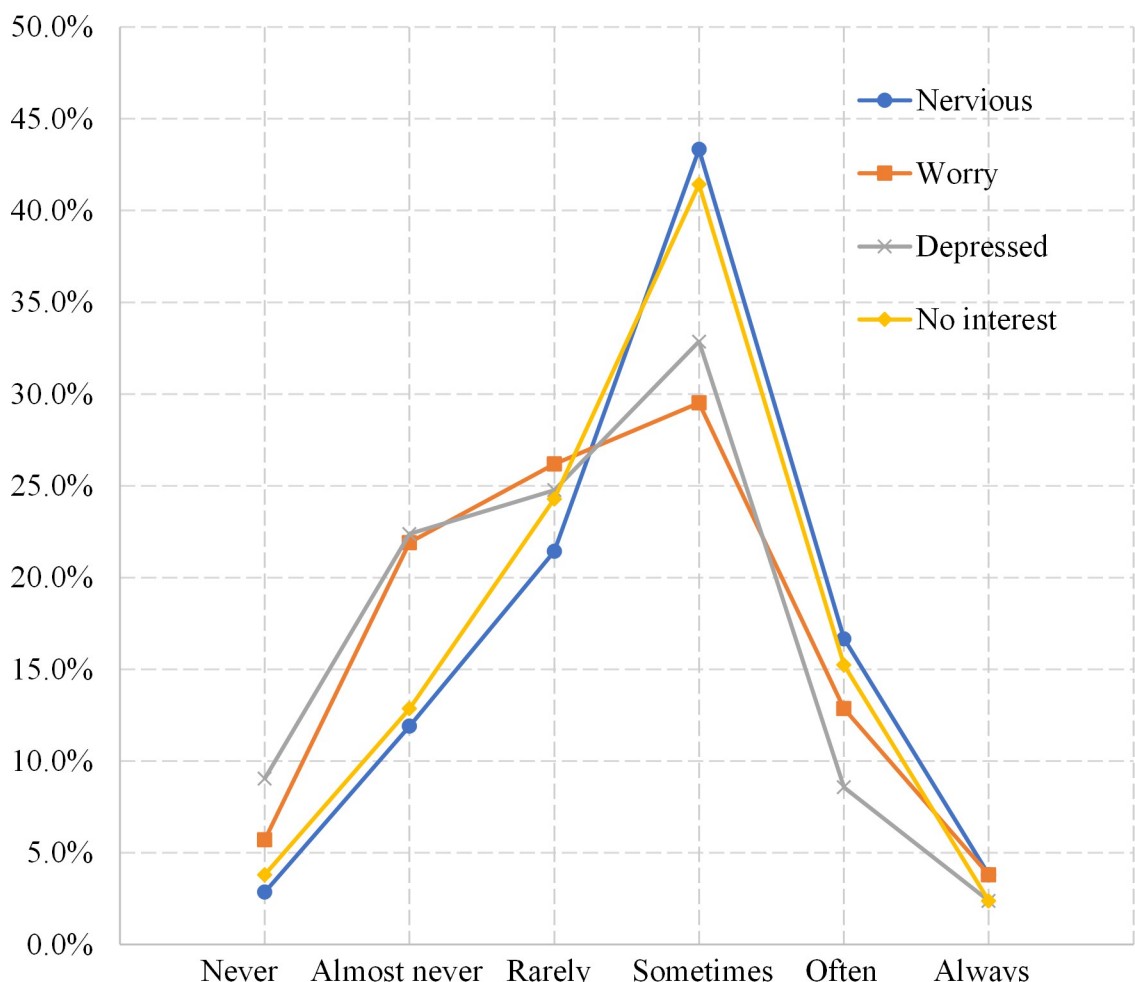

**Fig 3. Distribution of mental health response items.**

approximately 1415 meters). We then obtained the locations of all facilities in these grids and then measured RB and MB exposures.

This study adopted two methods to measure participants' exposures to environmental factors: the RB and MB methods. Each participant's RB exposures to environmental factors were measured by a series of circular or ring buffers around his/her residential location since it has been shown that the ring buffer could best capture the effects of environmental factors in people's residential neighborhoods on their health outcomes [59]. This research created circular/ring buffers with distances from 50m to 500m with 50 m intervals (50m, 50m-100m, 100m-150m, 150m-200m, 200m-250m, 250m-300m, 300m-350m, 350m-400m, 400m-450m, 450m-500m) [60], calculated RB environmental factors in these rings, and conducted a sensitivity analysis with environmental factors in each of the circular/ring buffers respectively. The results are similar with a slight difference. Finally, this study selected the ring buffer distance which has the highest explanatory power in the final model. Each participant's MB environmental factors were calculated with a similar method, except that we used GPS points in the two survey days as input to generate circular/ring buffers. To reduce the computational intensity and standardize the measurement, the GPS points were integrated into 1-minute intervals to obtain the MB exposures. We calculated the MB and RB transportation land-use density,

green space density and open space density, which is the percentage of the area of a specific land-use type in the total area of a ring buffer.

The population data were obtained from the HK Census and Statistics Department [61]. The areal unit for these data is the Street Block (SB) for urban areas or the Village Cluster (VC) in rural areas. The population density of each street block was calculated by dividing its population by its area (/km2). Note that MB population density cannot reveal real-time population density. Therefore, the population density was included only when using the RB method, which is the population density of the street block where each participant's home is located.

## 2.3 Analytical methods

We first applied linear regression to examine the association between participants' MB air pollution exposure and perceived air pollution. In this part, perceived air quality is the dependent variable, while MB air pollution exposure, MB/RB environmental factor exposures, socioeconomic status, and the mental health score are the independent variables. Since the dependent variable is normally distributed, as shown in Table 1 and verified by a Shapiro-Wilks test, this part uses linear regression to explore the association between the dependent variable and the independent variables. Model 1 uses RB environmental factors and Model 2 uses MB environmental factors.

We also explored whether people could perceive air quality relatively well or not. To achieve this, we classify actual (objectively measured) air quality and perceived air quality into two groups respectively. Actual air quality is divided into two groups (low: 3.208μg/m$^3$ - 15.536μg/m$^3$, high: 15.608μg/m$^3$-38.834μg/m$^3$) based on using K-means clustering given that people's sense and comparison of air quality is embedded in the actual air quality contexts in Hong Kong and official air quality guidelines (50μg/m$^3$) are too coarse to create balanced sub-groups [62]. Perceptions of air quality are classified into two groups (perceptions of air quality "Not at all", "Not serious", "Not very serious" as good, and "A little serious", "Serious", "Very serious" as poor). Therefore, the whole sample can be grouped into four types like a four-quadrant: (1) low air pollution concentrations and perceptions of air quality as good (accurate estimation: low-low), (2) high air pollution concentrations and perceptions of air quality as poor (accurate estimation: high-high), (3) low air pollution concentrations and perceptions of air quality as poor (overestimation: low-high), and (4) high air pollution concentrations and perception of air quality as good (underestimation: high-low) (Fig 4). Since these four groups have no numerical or ranking order among each other, we use multinomial logistic regression to examine which factors are prominent in each of the four groups. This part also includes two models, where Model 3 uses RB environmental factors and Model 4 uses MB environmental factors. All independent variables in the regression models are tested with multicollinearity diagnostics and the Variance Inflation Factor (VIF) is less than 10.

## 2.4. Participants' characteristics

The characteristics of the participants are shown in Table 2. We have 210 participants in total and 105 in each community. When compared to data from the 2016 Census, our sample closely mirrors the population demographics of SSP and TSW. Specifically, the gender proportions in both SSP and TSW have a high similarity with that in the census. There is a lower percentage of females in SSP compared to census data and there is a slightly higher percentage of female participants in TSW. For the age groups, the age group 25–44 in SSP is overrepresented and the age group 45–64 is underrepresented. In TSW, there are higher percentages of people in the age groups 18–24 and 25–44 than those in the 2016 Census and a smaller share of people in the age group 45–64 in our sample than those in the 2016 Census. The monthly household

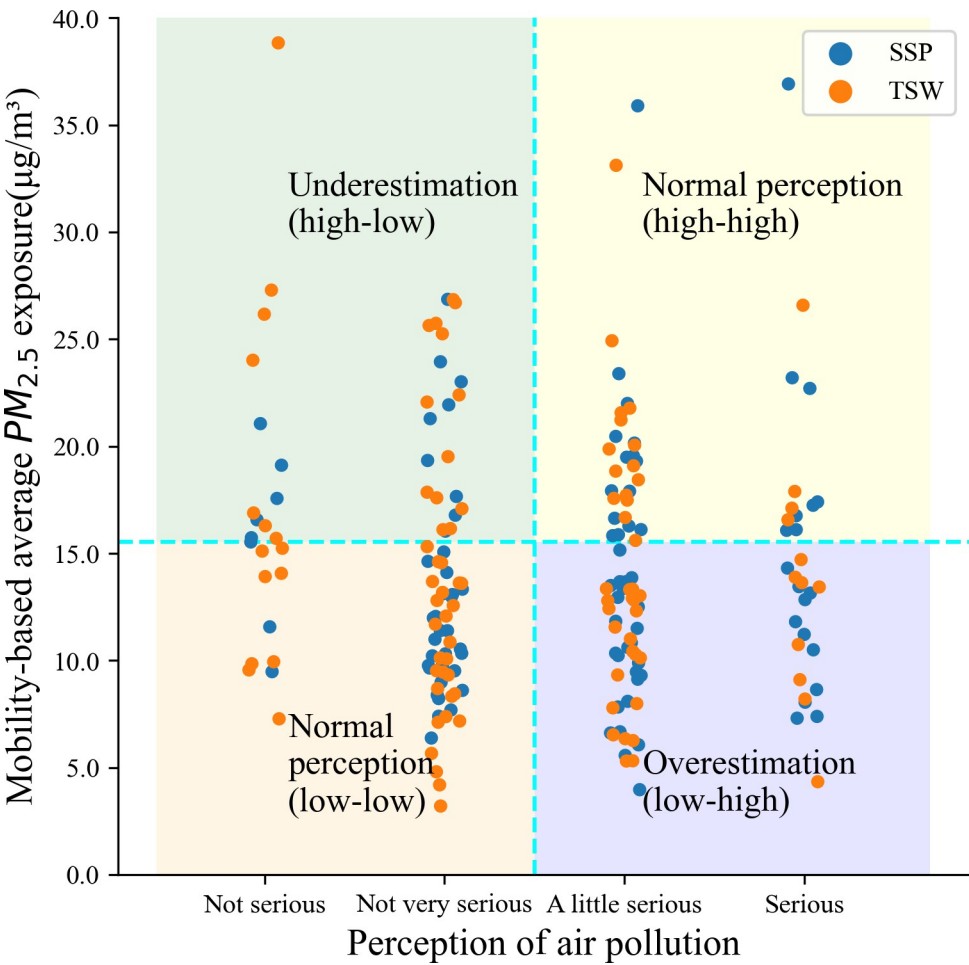

**Fig 4. Illustration of perception mismatch groups.**

income of the participants was divided into 7 groups: less than HK$9999, HK$10000–19999, HK$20000–29999, HK$30000–39999, HK$40000–59999, HK$60000–79999, and more than HK$80000, and they are coded from 1 to 7 respectively. To make a comparison with census income data, the income groups are reclassified into 4 groups as shown in Table 2. In the regression models, income is still represented as 7 groups. Income group 1 is underrepresented in SSP and highly underrepresented in TSW. Income groups 3 and 4 in both SSP and SSP are overrepresented. Higher percentages of the participants in SSP are employed and unemployed and a lower percentage of the participants are students when compared to census data. In TSW, the percentages of employed people and unemployed people among the participants are smaller than those in census data and students' share is larger than that in census data.

In the whole sample (Table 2), there are more females, people in the age group 25–44, participants with an income range of HK$20000–39999 and employed participants. Education levels 1 and 2 and education levels 5 and 6 have a large share of the participants. Half of the participants are single. Most of them are renters who rent their dwelling units (instead of being owner-occupiers), live in space smaller than 100 ft², with a family size smaller than or equal to 4. Most participants exercise for less than 3 days a week.

These sociodemographic attributes are included as the independent variables in the regression models. Gender, employment status, marital status, and homeownership are binary

**Table 2. Descriptive statistics of participants' characteristics.**

| Personal attributes | Code | Percent | SSP | | TSW | |
|---|---|---|---|---|---|---|
| | | | Sample | Census | Sample | Census |
| **Gender** | | | | | | |
| Male | 0 | 45.2% | 43.8% | 46.7% | 46.7% | 46.3% |
| Female | 1 | 54.8% | 56.2% | 53.3% | 53.3% | 53.7% |
| **Age** | | | | | | |
| 18–24 | - | 18.6% | 15.2% | 14.3% | 21.9% | 15.7% |
| 25–44 | - | 48.1% | 47.6% | 42.1% | 48.6% | 38.7% |
| 45–64 | - | 33.3% | 37.1% | 43.6% | 29.5% | 45.6% |
| **Household Income (HKD)** | | | | | | |
| < 9999 | 1 | 9.0% | 11.4% | 23.1% | 6.7% | 19.7% |
| 10000–19999 | 2 | 29.0% | 35.2% | 31.7% | 22.9% | 25.8% |
| 20000–39999 | 3 | 37.1% | 31.4% | 27.3% | 42.9% | 34.4% |
| >40000 | 4 | 24.8% | 21.9% | 17.8% | 27.6% | 20.0% |
| **Employment status** | | | | | | |
| Employed | 0 | 75.7% | 78.1% | 74.5% | 73.3% | 78.6% |
| Student | 1 | 11.0% | 8.6% | 14.3% | 13.3% | 6.7% |
| Unemployed | 1 | 13.3% | 13.3% | 11.2% | 13.3% | 14.8% |
| **Education** | | | | | | |
| Primary and below | 1 | 2.4% | | 1.9% | | 2.9% |
| Secondary | 2 | 33.8% | | 35.2% | | 32.4% |
| Post-secondary: Diploma/ Certificate | 3 | 8.6% | | 11.4% | | 5.7% |
| Post-secondary: Sub-degree course | 4 | 3.8% | | 4.8% | | 2.9% |
| Post-secondary: Degree course | 5 | 40.0% | | 36.2% | | 43.8% |
| Master and above | 6 | 11.4% | | 10.5% | | 12.4% |
| **Marital Status** | | | | | | |
| Never Married | 0 | 52.9% | | 49.5% | | 56.2% |
| Divorced | 1 | 8.1% | | 7.6% | | 8.6% |
| Married | 1 | 36.7% | | 40.0% | | 33.3% |
| Widowed | 1 | 2.4% | | 2.9% | | 1.9% |
| Separated | 1 | 0.0% | | | | |

variables coded as 0 and 1. The following variables are continuous: age ranging from 18 to 64, respiratory symptoms ranging from 3 (1×3 response items) to 18 (6×3 response items), physical exercise days in one week ranging from 0 to 7, mental health symptoms ranging from 4 (1×4 response items) to 24 (6×4 response items). Household income is an ordered categorical variable, but for convenience, we process it as a continuous variable and code it with an integer from 1 to 7, given that treating ordered categorical variables as continuous variables is reasonable when the classes are larger than 5 [63]. Education level is an ordered categorical variable, and we also code it with an integer from 1 to 6. Other independent variables are included in S1 Table.

## 3 Results

### 3.1. Descriptive analysis of variables

**3.1.1. Comparison between RB factors and MB factors.** Table 3 provides the basic statistics of the four environmental factors based on the RB and MB methods. Note that all the MB mean values are higher than the RB mean values. The maximum value of MB transportation

**Table 3. Description of residence-based (RB) and mobility-based (MB) environmental factors and significance levels of the Mann-Whitney U test.**

| Environmental exposures | | ring buffer | Mean | Std. | Min | Max | Sig. |
|---|---|---|---|---|---|---|---|
| Transportation land-use density | RB | 100m-150m | 3.10E-03 | 1.20E-03 | 2.30E-04 | 5.70E-03 | <0.001 |
| | MB | 350m-400m | 2.70E-01 | 6.30E-02 | 7.30E-02 | 4.50E-01 | |
| Green space density | RB | 100m-150m | 1.30E-04 | 3.80E-04 | 0.00E+00 | 2.90E-03 | <0.001 |
| | MB | 350m-400m | 8.30E-02 | 6.70E-02 | 1.50E-03 | 3.50E-01 | |
| Open space density | RB | 100m-150m | 6.30E-04 | 8.50E-04 | 0.00E+00 | 4.40E-03 | <0.001 |
| | MB | 350m-400m | 1.00E-01 | 6.10E-02 | 5.60E-03 | 3.40E-01 | |
| Facility density | RB | 100m-150m | 1.70E-04 | 1.50E-04 | 0.00E+00 | 7.60E-04 | 0.021 |
| | MB | 350m-400m | 1.30E-04 | 7.40E-05 | 1.20E-05 | 3.10E-04 | |
| Population density | RB | - | 1.30E+05 | 6.60E+04 | 6.00E+03 | 3.50E+05 | - |

land-use density is 100 times larger than that of RB transportation land-use density. The minimum, maximum and mean values of facility density based on the two measurements are similar. This result indicates that the daily mobility of the participants tends to increase their exposure to the four environmental factors.

We also conducted a nonparametric Mann-Whitney U test to examine whether the RB and MB measurements of these four environmental factors are significantly different. The results indicate that all environmental factors in each pair of RB and MB factors are significantly different from each other, suggesting the existence of the UGCoP. The results reveal that individual exposures to environmental factors may be different when different contextual areas (i.e., RB or MB) are used to derive them, and RB exposure cannot capture much of people's exposure in their activity spaces, indicating the necessity of using methods that can obtain more accurate results.

**3.1.2. Exposure disparity in groups.** In this subsection, we used ANOVA and box plots to show the variance of air pollution exposure between gender, age, income, and employment groups. Shapiro-Wilks test and Levene's test were conducted and passed before performing the ANOVA. Table 4 shows that there is a significant exposure disparity between different employment statuses and moderate exposure disparity among different income groups. Fig 5 also supports the results from Table 4 that exposures between different age groups and gender groups do not vary much. However, there is an obvious high exposure in the low-income group (<HK$9999) and the unemployed group.

## 3.2. Influencing factors of people's perceived air pollution

Linear regression is applied to study the association between participants' MB air pollution exposure and perceived air pollution (Table 5). Model 1 has the highest $R^2$ when the ring buffer is between 100m-150m. Model 2 has the highest $R^2$ when the ring buffer is between 350m-400m. The adjusted $R^2$ of the two regression models are 0.163 for the RB model and 0.125 for the MB model. The two models found that there is no significant association between air pollution exposure and perceived air quality. In the RB model, the coefficient for $PM_{2.5}$ is negative. In the MB model, the coefficient is positive.

Female participants tend to have positive assessments of the air quality in their surrounding environments when compared to male participants (Model 1). In Model 2, the difference

**Table 4. ANOVA analysis for four factors.**

| | Gender | Age | Income | Employment status |
|---|---|---|---|---|
| Anova sig | 0.534 | 0.204 | 0.079 | 0.008 |

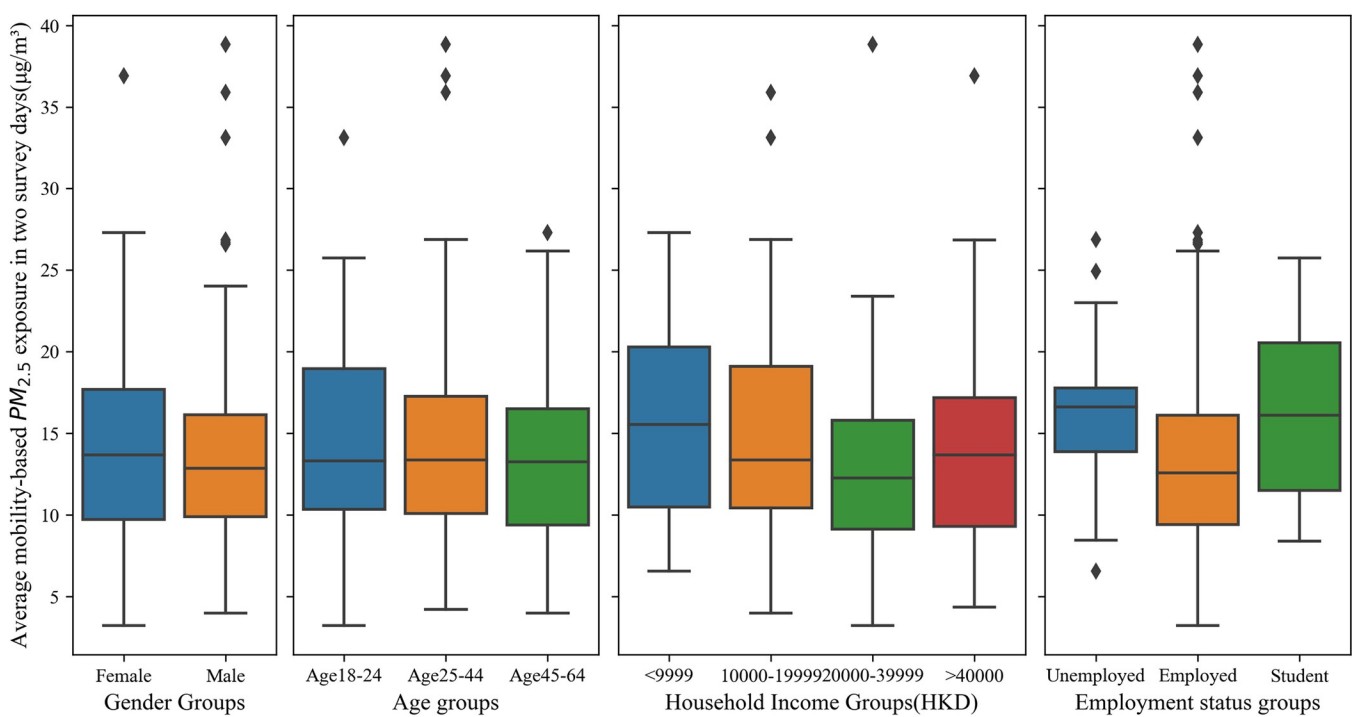

**Fig 5. Boxplot for exposure in sub-groups.**

between men's and women's assessments is not as significant as that in Model 1 and the absolute value of that coefficient in Mode 2 (1.718) is lower than that in Model 1 (2.016). Older people give a good assessment of the air quality at the places they visited in a typical week and the effect is stronger in Model 2 than that in Model 1. Compared to employed people, students have a higher chance to give positive air quality assessments (Model 2), and this effect is stronger than that in Model 1. People suffering from more serious mental health symptoms tend to have more negative air quality assessments (Models 1 and 2). Widowed people tend to have negative assessments of air quality in Model 2, with a higher coefficient for Model 2 when compared to that in Model 1, at a significance level of 0.1. People whose daily movements are surrounded by a higher density of MB transportation land use tend to have positive assessments of air quality (Model 2). This effect is not significant in Model 1. Higher RB green space density is associated with negative assessments of air quality (Model 1). The effect is weaker in Model 2. Facility density is associated with perceived air pollution (p< 0.1) (Model 1), and the effect is stronger in Model 1 than in Model 2.

## 3.3. Influencing factors of the mismatch between perceived air quality and air pollution exposure

This subsection explores the relationships between the mismatch in air pollution perceptions and actual air pollution and the influencing factors using multinomial logistic regression. In multinomial logistic regression, one group is selected as the reference group so that the regression results can be interpreted as the relative significance compared to the reference group. The accurate perception group (high-high) (i.e., no mismatch between actual and perceived air quality: negative assessments of air pollution and high exposure) is selected as the reference group and the regression results can be interpreted in relation to the reference group. To

**Table 5. Regression results of perceived air pollution (Model 1: Residence-based method (RB) and Model 2: Mobility-based (MB) method).**

| Variables | M1 RB (Ring buffer: 100m-150m) | | M2 MB (Ring buffer: 350m-400m) | |
|---|---|---|---|---|
| | Coef. | t value | Coef. | t value |
| (Intercept) | 3.397* | 2.121 | 3.392* | 2.153 |
| $PM_{2.5}$ | -0.005 | -0.446 | 0.002 | 0.171 |
| Gender (ref.: Male) | | | | |
| Female | -0.290* | -2.016 | -0.235. | -1.718 |
| Neighborhood (ref.: SSP) | | | | |
| TSW | -0.259 | -1.397 | -0.081 | -0.476 |
| Age | -0.013 | -1.373 | -0.022* | -2.555 |
| Education level | 0.053 | 0.861 | 0.037 | 0.618 |
| Marital status (ref.: Never married) | | | | |
| Divorced | 0.115 | 0.399 | 0.235 | 0.869 |
| Married | 0.222 | 1.109 | 0.302 | 1.551 |
| Widowed | 0.703 | 1.5 | 0.842. | 1.739 |
| Household income | 0.044 | 0.768 | 0.012 | 0.228 |
| Employment status (ref.: Employed) | | | | |
| Student | -0.16 | -0.714 | -0.432* | -2.016 |
| Unemployed | 0 | 0.002 | -0.02 | -0.103 |
| Family member | 0.021 | 0.33 | -0.034 | -0.55 |
| House Ownership (ref.: rent) | | | | |
| Own without mortgage | 0.061 | 0.287 | -0.07 | -0.377 |
| Own with mortgage | 0.058 | 0.29 | -0.163 | -0.905 |
| Living space | 0.013 | 0.133 | 0.097 | 1.116 |
| Physical exercise | 0.037 | 1.025 | 0.028 | 0.856 |
| Relative humidity | -0.006 | -0.796 | -0.014* | -2.18 |
| Temperature | -0.008 | -0.46 | -0.003 | -0.216 |
| Population density | 0 | -0.708 | 0 | -0.449 |
| Transportation land-use density | 89.6 | 1.414 | 2.593* | 2.015 |
| Green space density | 351.500* | 2.03 | 1.778. | 1.668 |
| Open space density | -3.824 | -1.415 | 0.002 | 0.069 |
| Facilities density | 1006.000. | 1.681 | 1177 | 0.961 |
| Respiratory symptoms | 0.038 | 1.5 | 0.025 | 1.124 |
| Mental disorder | 0.059*** | 3.63 | 0.051** | 3.231 |
| $R^2$ | 0.298 | | 0.237 | |
| Adjusted $R^2$ | 0.163 | | 0.125 | |
| Model's P-value | 0.002 | | 0.002 | |

Signif. codes: '***': 0.001

'**': 0.01

'*': 0.05; '.': 0.1; ' ': 1

obtain a valid model, we include independent variables that significantly influence perceived air quality in Models 1 and 2. Thus, the independent variables in this analysis include age, gender, community (i.e., SSP or TSW), employment status, living space, respiratory symptoms, relative humidity, mental health symptoms and the four RB and MB environmental factors. Model 3 is an RB model and Model 4 is an MB model (Table 6). The pseudo $R^2$ (Cox and Snell) in the two models are 0.398 and 0.331, showing relatively good predictions.

Specifically, the regression coefficients indicate which predictors significantly discriminate between participants whose perceived levels of air pollution are higher than, lower than, or

similar to actual air pollution levels. The first set of coefficients in Model 3 and Model 4 represents comparisons between participants who have accurate perceptions (high-high) and those who underestimate air quality (high-low). We found that age is associated with the underestimation group (Model 4), meaning that older people tend to have a higher chance to have positive assessments of air pollution while they are exposed to higher levels of air pollution. This effect is not significant in Model 3. Besides, people whose MB open space exposure is also positively related to the underestimation group (Model 4), meaning that people whose activity spaces have a higher density of open space tend to have a higher chance to underestimate air pollution when compared to the normal perception group. This association is not significant in Model 3. Mental health symptoms are negatively associated with the underestimation

**Table 6. Logistic regression of perception mismatch with normal perception (high-high) as the reference group (Model 3: Residence-based method (RB) and Model 4: Mobility-based (MB) method).**

| | | M 3 RB ring buffer 100m-150m | | M 4 MB buffer 350m-400m | |
|---|---|---|---|---|---|
| **Group** | **Variables** | **B** | **Sig.** | **B** | **Sig.** |
| Underestimation group | Intercept | -0.214 | 0.954 | -5.065 | 0.141 |
| | Gender(ref.:Male) | | | | |
| | Female | 0.057 | 0.932 | 0.458 | 0.405 |
| | Neighborhood(ref.:SSP) | | | | |
| | TSW | 1.327 | 0.128 | 0.498 | 0.495 |
| | Age | 0.023 | 0.452 | 0.049 | 0.06 |
| | Employment status(ref.:Employed) | | | | |
| | Student | 0.18 | 0.859 | 0.584 | 0.472 |
| | Unemployed | 0.683 | 0.412 | -0.017 | 0.981 |
| | Living space | -0.32 | 0.389 | 0.083 | 0.8 |
| | Relative humidity | 0.039 | 0.372 | 0.062 | 0.105 |
| | Transportation land-use density | -54.702 | 0.855 | -0.575 | 0.918 |
| | Green space density | 382.369 | 0.605 | 3.336 | 0.471 |
| | Open space density | 21.004 | 0.114 | 0.284 | 0.072 |
| | Facilities density | 142.746 | 0.957 | 789.622 | 0.866 |
| | Respiratory symptoms | -0.073 | 0.537 | -0.119 | 0.209 |
| | Mental disorder | -0.213 | 0.011 | -0.143 | 0.044 |
| Normal estimation group (low-low) | Intercept | 5.241 | 0.071 | 1.046 | 0.674 |
| | Gender(ref.:Male) | | | | |
| | Female | -0.047 | 0.932 | 0.17 | 0.712 |
| | Neighborhood(ref.:SSP) | | | | |
| | TSW | 0.314 | 0.66 | -0.506 | 0.405 |
| | Age | 0.008 | 0.758 | 0.032 | 0.149 |
| | Employment status(ref.:Employed) | | | | |
| | Student | -0.368 | 0.649 | -0.255 | 0.71 |
| | Unemployed | -1.272 | 0.123 | -1.078 | 0.111 |
| | Living space | -0.103 | 0.746 | 0.28 | 0.314 |
| | Relative humidity | -0.013 | 0.667 | 0.01 | 0.688 |
| | Transportation land-use density | -598.373 | 0.019 | -2.464 | 0.582 |
| | Green space density | -548.599 | 0.493 | -3.018 | 0.437 |
| | Open space density | 0.336 | 0.976 | 0.159 | 0.25 |
| | Facilities density | -2624.455 | 0.261 | -7112.959 | 0.105 |
| | Respiratory symptoms | 0.032 | 0.744 | 0.012 | 0.873 |
| | Mental disorder | -0.108 | 0.122 | -0.115 | 0.06 |

*(Continued)*

**Table 6.** (Continued)

| Group | Variables | M 3 RB ring buffer 100m-150m | | M 4 MB buffer 350m-400m | |
|---|---|---|---|---|---|
| | | B | Sig. | B | Sig. |
| Overestimation group | Intercept | 3.507 | 0.205 | -1.152 | 0.643 |
| | Gender(ref.:Male) | | | | |
| | Female | -1.102 | 0.037 | -0.491 | 0.276 |
| | Neighborhood(ref.:SSP) | | | | |
| | TSW | 0.169 | 0.812 | -0.264 | 0.66 |
| | Age | 0.003 | 0.9 | 0.021 | 0.331 |
| | Employment status(ref.:Employed) | | | | |
| | Student | -1.256 | 0.142 | -1.343 | 0.076 |
| | Unemployed | -1.011 | 0.208 | -1.342 | 0.061 |
| | Living space | 0 | 0.999 | 0.386 | 0.161 |
| | Relative humidity | -0.034 | 0.235 | -0.021 | 0.37 |
| | Transportation land-use density | -149.144 | 0.56 | 4.492 | 0.323 |
| | Green space density | 416.888 | 0.533 | 2.659 | 0.489 |
| | Open space density | 2.267 | 0.84 | 0.202 | 0.161 |
| | Facilities density | -1603.743 | 0.446 | -2290.222 | 0.577 |
| | Respiratory symptoms | 0.101 | 0.287 | 0.019 | 0.812 |
| | Mental disorder | -0.011 | 0.867 | -0.012 | 0.834 |
| Pseudo R-Square | Cox and Snell | 0.398 | | 0.331 | |
| | Nagelkerke | 0.425 | | 0.353 | |
| | McFadden | 0.183 | | 0.145 | |

group, indicating that people with more mental health symptoms have a larger chance to perceive air pollution negatively when they are exposed to poor air quality.

The second set of coefficients shows the factors influencing participants' accurate perceptions (low-low). Participants' RB transportation land-use density is negatively related to accurate perceptions (low-low) (Model 3), meaning that people whose residential neighborhoods have a higher density of transportation land use tend to have a higher chance to have accurate perceptions (high-high), and they tend to give negative assessments of air quality when exposed to have levels of high air pollution. This effect is not significant in Model 4. Mental health symptoms are negatively associated with accurate perceptions (low-low) (Model 4). The third set of coefficients shows the factors that influence the overestimation group. Compared to employed people, students and unemployed people are negatively associated with the overestimation group (Model 4), meaning that they are more positively related to the accurate estimation group (high-high). This indicates that compared to employed people, students and unemployed people tend to have a higher chance to have higher exposure and negative assessments of air quality. In Model 3, the effect direction is negative but less significant. Female participants are negatively associated with the overestimation group compared to male participants, meaning that they are positively related to the normal perception (high-high) group in the RB model.

## 4 Discussion and conclusion

We first used RB and MB approaches to assess people's multiple environmental exposures. Using separate regression models, we found that using different approaches can generate different outcomes in the associations between people's perceived air pollution with their multiple environmental exposures and socio-demographic attributes. Specifically, we found that

being older, a student and female, and relative humidity are positively related to perceptions of good air quality, while transportation land-use density, green space density, and mental disorder are positively related to a negative assessment of air quality in the MB model. When exploring our second objective of the mismatch between air pollution exposure and perception of air pollution, we found that using MB environmental factors and RB environmental factors could lead to different results. We found that older people and people who are exposed to higher MB open space density tend to underestimate air pollution, while people with severer mental disorders and unemployed people tend to have accurate perceptions of air pollution (high exposure and perceptions of poor air quality).

Models using RB and MB exposures have different explanatory powers and the coefficients and significance of each independent variable are also different. This study thus contributes to the study of perception of air pollution by showing how the UGCoP might affect the influence of environmental factors. Our study also conducted sensitivity analysis (S2 and S3 Tables). For the same ring buffer distance, we found that the RB $R^2$ and the Adjusted $R^2$ are larger than the MB $R^2$ and the Adjusted $R^2$ when the ring buffer distances are 100m-150m, 200m-250m, 300-350m, 450m-500m, and are smaller than the MB's $R^2$ and the Adjusted $R^2$ when the ring buffer distances are 150m-200m, 250m-300m, 350m-400m, 400m-450m. The RB models are not significant when the buffer distances are 50m and 50m-100m. Thus, these results showed that using RB and MB exposures could generate different explanatory powers for the dependent variable. However, caution should be paid to the RB and MB measurements and ring buffer distances when exploring environmental effects on perception of air pollution.

Comparing the largest $R^2$ and Adjusted $R^2$ of the RB models and the MB models, we found that the MB environmental factors have smaller explanatory power compared to the RB environmental factors (i.e., the $R^2$ in Model 1 is larger than that of Model 2, and the $R^2$ in Model 3 is larger than that of Model 4). This result indicates that influences of the RB and MB environmental factors are different over different spatial ranges. Besides, a previous study suggests that MB environmental exposures tend toward the mean value when the corresponding RB environmental exposures are much lower or higher than the mean exposure [64]. In other words, using the RB approach to measure individual exposures to environmental factors could overestimate or underestimate the environmental impact on people's perceived air pollution because it ignores the confounding effect of neighborhood effect averaging that arises from human daily mobility [27, 64].

Using MB real-time exposure, this study found no association between actual air pollution exposure and people's perception of air pollution. One possible explanation is that the actual air pollution in Hong Kong is relatively low. The 24-hour average limit of $PM_{2.5}$ in air quality guidelines in Hong Kong is 50μg/m$^3$ [62], and the maximum $PM_{2.5}$ concentration in our samples (48-hour average) is below 40μg/m$^3$. Besides, low levels of air pollution may lead to high visibility and less detectable odor from pollutants than high levels of air pollution. The average score of perceived air pollution in our study is 3.560, which represents a perception of slightly poor air quality. People's perception of air pollution, depending largely on sensory clues, might not recognize the risk when the clues are not obvious, such as carbon monoxide [6, 11]. However, what we should bear in mind is that continuous low levels of air pollution could produce crucial impacts on human health [65]. Hence, our study also implies that people might underestimate the harmful impact of air pollution on their health due to the continuously low level of air pollution exposure.

This study also identified socio-demographic groups that might have high exposure but underestimate air pollution and examined how the UGCoP could influence such findings. Gender difference in perceived air quality and perception mismatch (overestimation) was observed only in the RB models but not the MB models. These findings indicate that RB models could potentially exaggerate gender differences in perceived air quality and perception mismatch (perhaps due to differences in daily mobility between men and women).

The effects of age on perceived air quality and the underestimation group are significant only in the MB models. This finding is in line with a previous study that younger respondents tend to perceive air quality worse than older respondents [14]. When assessing the difference in perception of air quality, our study found that students tend to give positive assessments of air pollution (p<0.05). However, when exploring the differences in perception mismatch, our study found that students and unemployed people have a higher chance to have the accurate perception (high exposure and negative assessment) (p<0.1). Comparing the significance value, we conclude that students have a large tendency to think air quality is good. In other words, our results suggest that students tend to underestimate their air pollution exposure. This conclusion is inconsistent with results from [14]. Besides, unemployed people have significantly higher exposure (Table 4) and think air pollution is poor (Model 4, p<0.1). This result might support the self-selection thesis of people with low income or without jobs staying in environmentally polluted areas and their counterparts affording high-quality environments [66, 67]. Their sense of poor living/working conditions and low income might be potential factors contributing to their negative and objective assessment of air pollution.

Higher transportation land-use density is related to negative assessments of air quality in the MB models, and accurate estimation (high-high) in the RB models. These findings suggest that using only RB models is not enough to capture the effects of transportation land-use density on air pollution perception and mismatch. Transportation contributes a substantial part of air pollution [41]. Our results corroborate the previous studies that transportation is not only related to air pollution but also associated with perceptions of poor air quality [15, 22]. The traffic environment, such as traffic volume, traffic facilities, and transportation land-use density, might make people stressed and increase their worry about air pollution. Therefore, targeted efforts aiming at reducing the traffic burden could help to reduce air pollution and mitigate people's concerns about air pollution.

Higher green space density is related to negative air quality assessments in the RB and MB models with a more significant effect in the RB models than the MB models, demonstrating that using RB models could overestimate the effect of green space density on air pollution exposure and perceived air quality. This finding is in contrast with people's impression that more green space is related to good air quality [6]. Perhaps it is due to the fact that although HK keeps 40% of land as green space, most green spaces are located in suburbs with large areas, while green spaces in urban areas are relatively small and have many people socializing and relaxing at these locations and with a higher visit frequency than suburban green spaces [34, 35]. This situation might lead to people's impression that frequently visited green spaces in urban areas are associated with poor air quality. Even so, we should acknowledge the importance of green space in reducing air pollution and its restorative effect on mental health [68].

Open space includes parks, stadiums, playgrounds, and recreational facilities [69]. Higher open space density is related to the underestimation group in the MB models, meaning that people at these places tend to have a higher exposure but tend to give positive assessments of air quality. People might perform physical exercise, entertain, and socialize at these parks or stadiums voluntarily. Therefore, this result might indicate that people would think air pollution is less serious when they are conducting activities whose benefits outweigh the costs [5].

Higher relative humidity is related to perceptions of good air quality in the MB model (Model 2). In Model 4, higher relative humidity is associated with the tendency to underestimate air pollution situation (p = 0.105). When conducting our survey from March to September, HK had entered into the rainy season, which means that relative humidity and rain volume are continuously higher than in other dry months. Rain would wash away air pollutants in high relative humidity conditions, helping to reduce air pollution concentration [70]. Therefore, we conclude that people do not underestimate air pollution in high relative

humidity conditions and people's perceptions of good air quality in high relative humidity conditions and rainy days are aligned with scientific findings.

Mental health symptoms are related to negative assessments of air quality in the RB and MB models and are associated with accurate estimations (high-high) in the MB models, having results similar to a previous study that better mental health was related to lower levels of perceived air pollution in China, Japan and South Korea [71]. Another study exploring the influencing factors of mental status found that it is the subjective perception of air pollution rather than the objective measure of air pollution that influences people's mental health [72]. Cognitive representation of people's environmental factors might play a fundamental role in mental health [68]. And mental health might contribute to the psychological processing of environmental factors and influence people's perception of air pollution.

Public perceptions of air pollution stem from what people can see, hear, smell, feel, or have suffered. Objective environmental factors are not perceived equally by different individuals. Individuals' subjective feelings about different environmental factors are crucial for assessing environmental quality and mental health [72]. Public perception in microenvironments can be a good dimension to assess the "livability" of a city [8].

These results have important implications. First, our study found that the association between environmental factors and perception of air pollution could be different when RB and MB measurements are used. Therefore, studies on mobility-dependent exposures need to recognize the UGCoP by considering people's daily mobility. Environmental health research with RB measurements neglects the spatial and temporal variations of mobility-dependent exposures, which may lead to inaccurate results. Second, ignoring mobility and failing to address the UGCoP might mislead policymakers when researchers explore the association between mobility-dependent environmental factors and perception of air pollution. For example, our study found that older people tend to underestimate air pollution only in the MB model, not in the RB model. Therefore, public air pollution education (such as the nature and magnitude of air pollution problems, emission sources, and related health risks) should be held, especially for older people who tend to underestimate air pollution. This could increase people's basic knowledge about air pollution, rectify the misunderstanding of air pollution, and enhance comprehension and cooperation among the public, environmental departments, and academic communities [73]. Without using the MB method, policies might lose focus and have low efficiency. Using the MB method, our study also found that transportation land-use density and open space density are related to people's perception of air pollution. Therefore, policymaking should consider the psychological, cognitive, and mental health effects of landscapes when planning our environment. Besides, people's perceptions of the environment are quite momentary and influenced by specific microclimates. Real-time and high-resolution spatial information on air pollution could be provided to guide people's daily activities and travel to reduce their frequency and duration at high air-pollution microenvironments.

This research has several strengths. First, the methods used in our study can be generalized to other cities and communities in Hong Kong. Our study has a well-designed approach in exploring the association and mismatch between air pollution exposure and perception of air pollution. In addition, our study used RB and MB environmental factors as independent variables in RB/MB models to examine whether the measurements could influence the association and mismatch between air pollution and perception of air pollution. Other researchers with similar goals could adopt our study design and methods. Further, as individual-level air pollution measurement becomes increasingly important for exploring exposure disparity, the uncertainty of air pollution exposure estimation methods, and the association between environmental risks and health outcomes, our survey method provides a basic framework for

researchers to utilize portable sensors to conduct research measuring real-time individual-level environmental exposure.

There are several limitations in our study. First of all, our study used DustTrak as a reference sensor and calibrated air pollution exposure based on its measurement. DustTrak is precise when measuring a wide range of particle matter and its $R^2$ of linear regression between static monitoring stations is higher than 0.9 [74]. However, DustTrak's performance decreases when RH is higher than 60% [74]. Our measurement period started from March to September, which is the summer and rainy season, with high RH in HK. Thus, the calibrated results might be not so precise in high RH days as in sunny days. Our further study would adopt other reference sensors and adopt RH-based calibrated method to improve accuracy. Even though with this limitation, our study result is still valid given that our previous study found that there was a high correlation between Airbeam-2 sensors' measurement of the same ambient air in different microenvironments [54]. This result indicates that our sensors' internal consistency is high. Airbeam-2's measurements can be used to explore the relationship between air pollution exposure and perception of air pollution, health outcomes, and other influencing factors since absolute readings of air pollution is not required. The absolute and precise measurement of air pollution is needed for research aiming at exploring the non-stationary effect of environmental factors on the outcome variables, such as the threshold effect and the value-range effect [75].

Second, our research could have limited representativeness due to the small sample size. Even though we considered population structure, there may be some self-selection bias due to participants' willingness to join this study. The small sample size could also lead to the small $R^2$ in our models and insignificant associations for most of variables. This could further limit our research findings' application in other study areas. We believe that with a larger sample size, models would show a larger $R^2$ and significant associations for more independent variables. Models with larger sample size could be generalized to larger study areas. With these limitations, our study findings are still valid since our sensitivity analysis showed stability of these results (S2 and S3 Tables). Future studies can consider other survey methods to improve the representativeness of the sample. Third, the survey was conducted in different seasons in HK, which may have seasonality effects [76]. To reduce this effect, this study included temperature and humidity in the models, taking the atmospheric conditions into account. Fourth, people's views about air quality may change over different environments where they conducted diverse activities. Therefore, future studies can explore how people's momentary experiences, activities and mental health status affect their assessments of air pollution. Fourth, people's view towards the environment is not only about what they think about the environmental quality but also reflects their deep-seated values and world-views and whether they have trust in governmental and individual efforts to improve the environment [5, 15]. Further studies can explore more dimensions of public perceptions of air pollution.

## Supporting information

**S1 Table. Descriptive statistics of participants' characteristics.**
(DOCX)

**S2 Table. Regression results of association between influencing factors and perception of air pollution using residence-based methods.**
(DOCX)

**S3 Table. Regression results of association between influencing factors and perception of air pollution using mobility-based methods.**
(DOCX)

## Acknowledgments

The authors would like to thank the participants for their participation in the study and the anonymous reviewers for their helpful comments.

## Author Contributions

**Conceptualization:** Wanying Song, Mei-Po Kwan, Jianwei Huang.

**Formal analysis:** Wanying Song.

**Funding acquisition:** Mei-Po Kwan.

**Methodology:** Wanying Song, Mei-Po Kwan, Jianwei Huang.

**Supervision:** Mei-Po Kwan.

**Writing – original draft:** Wanying Song, Mei-Po Kwan, Jianwei Huang.

**Writing – review & editing:** Wanying Song, Mei-Po Kwan, Jianwei Huang.

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
