## [Decision Letter · Decision Letter 0]

3 Apr 2023

PONE-D-23-04918Assessment of Air Pollution and Air Quality Perception Mismatch Using Mobility-based Real-time ExposurePLOS ONE

Dear Dr. Kwan,

Thank you for submitting your manuscript to PLOS ONE. After careful consideration, we feel that it has merit but does not fully meet PLOS ONE’s publication criteria as it currently stands. Therefore, we invite you to submit a revised version of the manuscript that addresses the points raised during the review process.

We look forward to receiving your revised manuscript.

Kind regards,

Tai Ming Wut

Academic Editor

PLOS ONE

Journal Requirements:

5. Please include a copy of Table 7 which you refer to in your text on page 21.

Reviewers' comments:

Reviewer's Responses to Questions

**Comments to the Author**

1. Is the manuscript technically sound, and do the data support the conclusions?

Reviewer #1: Partly

Reviewer #2: Yes

2. Has the statistical analysis been performed appropriately and rigorously? 

Reviewer #1: Yes

Reviewer #2: Yes

3. Have the authors made all data underlying the findings in their manuscript fully available?

Reviewer #1: No

Reviewer #2: Yes

4. Is the manuscript presented in an intelligible fashion and written in standard English?

Reviewer #1: Yes

Reviewer #2: Yes

5. Review Comments to the Author

Reviewer #1: The manuscript titled “Assessment of Air Pollution and Air Quality Perception Mismatch Using Mobility-based Real-time Exposure” is a well-written and well-organized paper. It presents data and analysis to explore the mismatching perceived exposure to air pollution and its influencing factors. The data collection includes real-world exposure measurement and surveys on selected population groups; the analysis involves different statistical models trying to stratify different effects. It is an overall well-designed study. However, there are some concerns regarding data calibration, data collection and model interpretations as stated below. Additional analysis and/or discussions are needed to make these points clear.

Specific comments:

1. For calibration, author claimed that DustTrak is a “high-accuracy” sensor and used it to calibrate AIRBEAM, the sensor used in the study. However, DustTrak used light scattering method to measure particles, which is heavily influenced by environmental conditions, such as humidity. Previous studies in Hong Kong reported overestimates by factors of 2-4 from PM measured by DustTrak compared to regulatory monitors. An uncalibrated DustTrak is an inaccurate ruler and may bring bias to the measurement results. Authors should re-evaluate the calibration and discuss possible impacts on the results and conclusion.

2. For data collection, exposure was measured for each participant based on their mobility (MB). There is no residential-based (RB) exposure measurement. However, the major models used in the data analysis were constructed in both ways. For models constructed in RB way, MB exposure is used, which making it difficult to compare and address UGCoP.

3. For model interpretation, the constructed models in Table 8 and Table 9 only explained small portion of variations in the dependent variable and the regression results are insignificant for most parameters. As such, I am not sure whether the interpretations are overstretched.

4. The ring buffer for MB is different from RB. The former is larger. How would this affect the comparison? Would the conclusion remain unchanged if the same ring buffer is used for both?

5. Figure 5, it would be better to arrange categories in order on the plot (e.g. Age 18-24, Age25-44, Age 45-64) for easier comparison and understanding.

Reviewer #2: This paper examines the discrepancy between perceived air pollution and actual pollution, as well as variations among different social groups. The study takes a mobility-based geographic context approach to address the challenge of investigating spatial pollution exposure using only static geographic space, such as residential areas. By doing so, it reduces potential biases and holds considerable scientific value.

There are followings suggestions:

Can you provide additional context or information about the dynamic approach that has been proposed on the relationship between environmental pollution, perceived pollution and health?

Whether the methods and results of the study be generalized to other cities or communities in Hong Kong?

When measuring actual pollution exposure, PM2.5 is measured by the average mean value of exposure for the two survey days (the two days include one weekday and one

weekend day) in this study. Would it be more accurate to consider weekdays and weekends separately? Given that weekdays make up five out of seven days of the week, could a weighting system be implemented to address this issue?

Since apply RB and MB approaches relying on different regression models to assess people's multiple environmental exposures, may produce divergent findings regarding the associations between people's perceived air pollution, their environmental exposures, and socio-demographic attributes. In the future, which model should be selected, and how can policy recommendations be effectively formulated?

6. PLOS authors have the option to publish the peer review history of their article (what does this mean?). If published, this will include your full peer review and any attached files.

Reviewer #1: No

Reviewer #2: No

---

## [Author Response · Author response to Decision Letter 0]

27 Apr 2023

Response to Reviewers’ comments

Dear Editor,

We appreciate your decision letter and the helpful feedback from the reviewers on our manuscript. We have carefully revised our paper to address their comments and have made many corrections to address their concerns. We have highlighted major revisions in red text for your convenience.

We are particularly grateful to you and the reviewers for your efforts in handling and reviewing our paper. 

The thoughtful comments provided by the reviewers have greatly assisted us in improving the paper. We look forward to hearing from you regarding our revised manuscript.

Best regards,

The Authors

Journal Requirements:

[Comment 1] Please ensure that your manuscript meets PLOS ONE's style requirements, including those for file naming. The PLOS ONE style templates can be found at 

(Response 1) Thanks for your comment. The authors have thoroughly checked and revised the formats of all elements in our manuscript according to guidelines. 

[Comment 2] We note that the grant information you provided in the ‘Funding Information’ and ‘Financial Disclosure’ sections do not match. 

(Response 2) Thanks for your checking. We have revised the grant information accordingly.

[Comment 3] Your ethics statement should only appear in the Methods section of your manuscript. If your ethics statement is written in any section besides the Methods, please move it to the Methods section and delete it from any other section. Please ensure that your ethics statement is included in your manuscript, as the ethics statement entered into the online submission form will not be published alongside your manuscript. 

(Response 3) Thanks for the comment. Our ethics statement was included in the Methods section. Besides, we checked the manuscript and delete the ethics statement at the end of manuscript.

(Page 10, Lines 197-199)

“The study protocol and survey instruments were reviewed and approved by the Survey and Behavioural Research Ethics Committee of the authors’ university, and informed consent was obtained from all participants”

 [Comment 4] We note that Figure 1 in your submission contain [map/satellite] images which may be copyrighted. All PLOS content is published under the Creative Commons Attribution License (CC BY 4.0), which means that the manuscript, images, and Supporting Information files will be freely available online, and any third party is permitted to access, download, copy, distribute, and use these materials in any way, even commercially, with proper attribution. For these reasons, we cannot publish previously copyrighted maps or satellite images created using proprietary data, such as Google software (Google Maps, Street View, and Earth). For more information, see our copyright guidelines: http://journals.plos.org/plosone/s/licenses-and-copyright.

(Response 4) Thanks for your suggestion. The data of land use classification was obtained from an online open source: Land Utilization in Hong Kong 2021( https://www.pland.gov.hk/pland_en/info_serv/open_data/landu/index.html#!)

The boundary of Sham Shui Po (SSP) and Tin Shui Wai (TSW) was obtained from another online open source data: Shapefile of DCCA boundary: 2016 Population By-census Statistics (By Small Street Block Group) (https://geodata.gov.hk/gs/view-dataset?uuid=ae957bd4-0067-407b-83c5-a930c253d7d1&sidx=0 )

The authors made the map themselves by using different colours to represent each land use type. 

[Comment 5] Please include a copy of Table 7 which you refer to in your text on page 21.

(Response 5) Thanks for your comment. We added Table 7 on page 22.

(Page 22, Line 376)

Table 7. ANOVA analysis for four factors.

 Gender Age Income Employment status

Anova sig 0.534 0.204 0.079 0.008

 

Reviewer(s)' Comments to Author:

Reviewer 1

Reviewer #1: The manuscript titled “Assessment of Air Pollution and Air Quality Perception Mismatch Using Mobility-based Real-time Exposure” is a well-written and well-organized paper. It presents data and analysis to explore the mismatching perceived exposure to air pollution and its influencing factors. The data collection includes real-world exposure measurement and surveys on selected population groups; the analysis involves different statistical models trying to stratify different effects. It is an overall well-designed study. However, there are some concerns regarding data calibration, data collection and model interpretations as stated below. Additional analysis and/or discussions are needed to make these points clear.

Specific comments:

[Comment 1] For calibration, author claimed that DustTrak is a “high-accuracy” sensor and used it to calibrate AIRBEAM, the sensor used in the study. However, DustTrak used light scattering method to measure particles, which is heavily influenced by environmental conditions, such as humidity. Previous studies in Hong Kong reported overestimates by factors of 2-4 from PM measured by DustTrak compared to regulatory monitors. An uncalibrated DustTrak is an inaccurate ruler and may bring bias to the measurement results. Authors should re-evaluate the calibration and discuss possible impacts on the results and conclusion.

(Response 1) Thanks for your suggestions. The authors have changed description of “DustTrak”. The author added the discussion of the bias caused by DustTrak in the discussion part. We also discussed the feasibility of using Airbeam-2 sensor in different research goals. 

(Page 10, Line 202-204)

“Since portable sensors may sacrifice accuracy for convenience, we calibrated the portable sensors using DustTrak as a reference sensor based on a model we previously established [50].”

(Page 35, Line 604-614)

“First of all, our study used DustTrak as a reference sensor and calibrated air pollution exposure based on its measurement. DustTrak is precise when measuring wide range of particle matters and its R2 of linear regression between static monitoring stations is higher than 0.9[70]. However, DustTrak’s performance decreases when RH is higher than 60% [70]. Our measurement period started from March to September, which is the summer and rainy season, with high RH in HK. Thus, the calibrated results might be not so precise in high RH days as in sunny days. Our further study would adopt other reference sensors and adopt RH-based calibrated method to improve accuracy. Even though with this limitation, our study result is still valid given that our previous study found that there was a high correlation between Airbeam-2 sensors’ measurement of the same air in different microenvironments [50]. This result indicates that our sensors’ internal consistency is high.”

(Page 35-35, Line 615-619)

“Airbeam-2’s measurements can be used to explore the relationship between air pollution exposure and perception of air pollution, health outcomes, and other influencing factors since absolute readings of air pollution is not required. Absolute and precise measurement of air pollution is needed for research aiming at exploring the non-stationarity effect of environmental factors on outcome variables, such as threshold effect, value-range effect [71].”

References:

50. Huang J, Kwan M-P, Cai J, Song W, Yu C, Kan Z, et al. Field Evaluation and Calibration of Low-Cost Air Pollution Sensors for Environmental Exposure Research. Sensors. 2022;22(6):2381.

70. Li Z, Che W, Lau AKH, Fung JCH, Lin C, Lu X. A feasible experimental framework for field calibration of portable light-scattering aerosol monitors: Case of TSI DustTrak. Environmental Pollution. 2019;255:113136. doi: https://doi.org/10.1016/j.envpol.2019.113136.

71. Kwan M-P. The stationarity bias in research on the environmental determinants of health. Health & Place. 2021;70:102609. doi: https://doi.org/10.1016/j.healthplace.2021.102609.

[Comment 2] For data collection, exposure was measured for each participant based on their mobility (MB). There is no residential-based (RB) exposure measurement. However, the major models used in the data analysis were constructed in both ways. For models constructed in RB way, MB exposure is used, which making it difficult to compare and address UGCoP.

(Response 2) Thanks for the comment. The study design proposed by reviewer is suitable for exploring the UGCoP in air pollution exposure. This question has been well examined in previous studies [1, 2]. Therefore, we do not include RB air pollution exposure in our RB models. We acknowledge the existence of the UGCoP when exploring the association between air pollution and perception of air pollution and addressed this problem by using only MB air pollution exposure in RB and MB models. Our study focuses on the UGCoP of transportation land-use density, green space density, open space density, and facilities density when exploring these factors’ associations with perception of air pollution. This study keeps MB exposure to air pollution, individual’s social economic status and health status as independent variables in RB and MB models, and keeps transportation land-use density, green space density, open space density, and facilities density different in geographic delineations. In this way, we would be able to know that the difference in explanatory power in RB and MB models was caused by the different RB and MB measurement of the four above-mentioned independent variables. By comparing the coefficients of RB and MB independent variables, we learn that using different geographic delineations to measure the mobility-dependent independent variables could generate different correlations between these independent variables and people’s perception of air pollution.

Our results show that the UGCoP exists when we examine the association between perception of air pollution and transportation land-use density, green space density, open space density, and facilities density. According to the associations between the four environmental factors and individual perception of air pollution with RB and MB delineation methods, environmental factors that is significantly correlated with perception of air pollution could be classified into two categories: (1) environmental factors that have similar relationships with perception of air pollution with RB and MB methods. For example, green space density is significantly and positively related to perception of air pollution; (2) environmental factors that have different relationships with perception of air pollution with RB and MB methods. For instance, transportation land-use density is only significantly associated with perception of air pollution in MB model (Model2). Facilities density is only significantly associated with perception of air pollution in RB model (Model 1). Age and open space density are significantly associated with underestimation group only in MB model (Model 4). Hence, using RB and MB environmental factors in different RB and MB models and keeping other independent variables the same, we can learn that the UGCoP could influence the results when we want to explore the influencing factors of perception of air pollution, and the association and mismatch between air pollution and perception of air pollution. 

References:

1. Ma J, Tao Y, Kwan M-P, Chai Y. Assessing mobility-based real-time air pollution exposure in space and time using smart sensors and GPS trajectories in Beijing. Annals of the American Association of Geographers. 2020;110(2):434-448.

2. Kim J, Kwan M-P. How Neighborhood Effect Averaging Might Affect Assessment of Individual Exposures to Air Pollution: A Study of Ozone Exposures in Los Angeles. Annals of the American Association of Geographers. 2021;111(1):121-140. doi: 10.1080/24694452.2020.1756208.

[Comment 3] For model interpretation, the constructed models in Table 8 and Table 9 only explained small portion of variations in the dependent variable and the regression results are insignificant for most parameters. As such, I am not sure whether the interpretations are overstretched.

(Response 3) Thanks for the comment. The authors added discussions of the small R2 and insignificant associations for most of the variables in the revised manuscript. We included sensitivity analysis of Model 1 and Model 2 (S 1-2 Tables) in the supplementary materials for review. We did not conduct sensitivity analysis for Model 3 and Model 4 because the best ring buffer is settled in Model 1 and Model 2. 

(Page 36, Line 620-628)

“Second, our research could have limit representativeness due to small sample size. Even though we considered population structure, there may be some self-selection bias due to participants’ willingness to join this study. Small sample size could also lead to small R2 in our models and insignificant associations for most of variables. This could further limit our research findings application in other study areas. We believe that with larger sample size, models would show a larger R2 and significant associations for more independent variables. Models with larger sample size could be generalized to a larger study area. With these limitations, our study findings are still valid since our sensitivity analysis showed stability of these results (S 1-2 Tables). Future studies can consider other survey methods to improve the representativeness of the sample.” 

[Comment 4] The ring buffer for MB is different from RB. The former is larger. How would this affect the comparison? Would the conclusion remain unchanged if the same ring buffer is used for both?

(Response 4) Thanks for your question. We used the ring buffer which can have the highest explanatory power in Model 1 and Model 2 separately. This means that mobility-based environmental factors and residence-based environmental factors have influences over different spatial ranges. For the RB model, the ring buffer of 100m-150m can best explain the perception of air pollution. For the MB model, the ring buffer is 350-400m. 

If we use the same ring buffer in the RB and MB models, the significantly associated variables stay almost the same, and the significance value of the association might be different. The explanation power of RB and MB models are also different. We added more discussion in the manuscript to clarify this.

(Page 29, Line 465-476)

“Models using RB and MB exposures have different explanatory powers and the coefficients and significance of each independent variable are also different. This study thus contributes to the study of perception of air pollution by showing how the UGCoP might affect the influence of environmental factors. Our study also conducted sensitivity analysis (S1-2 Tables). For the same ring buffer distance, our study found that RB R2 and Adjusted R2 are larger than MB R2 and Adjusted R2 when the ring buffer distances are 100m-150m, 200m-250m, 300-350m, 450m-500m, and are smaller than MB’s R2 and Adjusted R2 when the ring buffer distances are 150m-200m, 250m-300m, 350m-400m, 400m-450m. RB’s models are not significant when the buffer distances are 50m and 50m-100m. Thus, these results showed that using RB and MB could generate different explanatory powers for dependent variable. Caution should be paid to the RB and MB measurements and ring buffer distances when exploring environmental effects on perception of air pollution.”

(Page 29-30, Line 477-486)

“Comparing the largest R2 and Adjusted R2 of RB models and MB models, we found that MB environmental factors have smaller explanatory power compared to the RB environmental factors (i.e., the R2 in Model 1 is larger than Model 2, and the R2 in Model 3 is larger than Model 4). This result indicates that RB and MB environmental factors have different influences in spatial ranges. Besides, a previous study suggests that MB environmental exposures tend toward the mean value when the corresponding RB environmental exposures are much lower or higher than the mean exposure [60]. In other words, using the RB approach to measure individual exposures to environmental factors could overestimate or underestimate the environmental impact on people’s perceived air pollution because it ignores the confounding effect of neighbourhood effect averaging that arises from human daily mobility [26, 60].”

[Comment 5] Figure 5, it would be better to arrange categories in order on the plot (e.g. Age 18-24, Age25-44, Age 45-64) for easier comparison and understanding.

(Response 5) Thanks for your suggestion. We have replotted Figure 5 to make it more readable. 

(Page 22, Line 377)

 

Reviewer 2

Reviewer #2: This paper examines the discrepancy between perceived air pollution and actual pollution, as well as variations among different social groups. The study takes a mobility-based geographic context approach to address the challenge of investigating spatial pollution exposure using only static geographic space, such as residential areas. By doing so, it reduces potential biases and holds considerable scientific value.

There are followings suggestions:

[Comment 1] Can you provide additional context or information about the dynamic approach that has been proposed on the relationship between environmental pollution, perceived pollution and health?

(Response 1) Thanks for reviewers’ comments. We added studies using dynamic approach to estimate air pollution in introduction part. 

(Page 6-7, Line 121-133)

“There are several methods to tackle with the potential bias caused by static air pollution measurements. The first and direct measurement method is measuring individual real time air pollution exposure using portable sensors. For example, Zhang et al. used RB and MB approaches to estimate environmental exposures and explored the association between environmental exposures and mental health [28]. She found that MB approach has a higher explanation power on mental health than RB approach. Another method to estimate individual MB exposure is using exposure equation based on people’s body weight, respiratory rates, indoor to outdoor air pollution concentrations, and time-activity patterns based on air pollution monitoring stations data [16]. Studies usually obtained hourly or daily air pollution monitoring station data and estimated air pollution in whole study area based on spatial interpolation, combined with satellite data, or air dispersion models [26, 29, 30]. A study estimating individual air pollution exposure by monitoring stations air pollution data and exposure equation found a correlation between individual exposure levels and risk perception of PM2.5[16].”

References:

16. Huang L, Rao C, van der Kuijp TJ, Bi J, Liu Y. A comparison of individual exposure, perception, and acceptable levels of PM2.5 with air pollution policy objectives in China. Environmental Research. 2017;157:78-86. doi: https://doi.org/10.1016/j.envres.2017.05.012.

26. Kim J, Kwan M-P. How Neighborhood Effect Averaging Might Affect Assessment of Individual Exposures to Air Pollution: A Study of Ozone Exposures in Los Angeles. Annals of the American Association of Geographers. 2021;111(1):121-140. doi: 10.1080/24694452.2020.1756208.

28. Zhang L, Zhou S, Kwan M-P, Shen M. Assessing individual environmental exposure derived from the spatiotemporal behavior context and its impacts on mental health. Health & Place. 2021;71:102655. doi: https://doi.org/10.1016/j.healthplace.2021.102655.

29. Knibbs LD, Hewson MG, Bechle MJ, Marshall JD, Barnett AG. A national satellite-based land-use regression model for air pollution exposure assessment in Australia. Environmental Research. 2014;135:204-211. doi: https://doi.org/10.1016/j.envres.2014.09.011.

30. Park YM. Assessing personal exposure to traffic-related air pollution using individual travel-activity diary data and an on-road source air dispersion model. Health & Place. 2020;63:102351.

[Comment 2] Whether the methods and results of the study be generalized to other cities or communities in Hong Kong?

(Response 2) Thanks for reviewer’s question. We think that the methods in our study can be generalized to other cities and communities in Hong Kong. We added more discussion on this in the revised manuscript. 

(Page 34-35, Line 593-603)

“This research has several strengths. First, the methods used in our study can be generalized to other cities and communities in Hong Kong. Our study has a well-designed approach in exploring the association and mismatch between air pollution exposure and perception of air pollution. Additionally, our study used RB and MB environmental factors as independent variables in RB/MB models to examine whether the measurements could influence the association and mismatch between air pollution and perception of air pollution. Other researchers with similar goals could adopt our study design and methods. Furthermore, as individual-level air pollution measurement becomes increasingly important for exploring exposure disparity, uncertainty of air pollution exposure estimation methods, and the association between environmental risks and health outcomes, our survey method provides a basic framework for researchers to utilize portable sensors to conduct research measuring real-time individual-level environmental exposure.”

However, the results may not be generalized to other communities in Hong Kong due to small sample size in our study. We add more discussion on this in the revised manuscript. 

(Page 36, Line 620-628)

“Second, our research could have limit representativeness due to small sample size. Even though we considered population structure, there may be some self-selection bias due to participants’ willingness to join this study. Small sample size could also lead to small R2 in our models and insignificant associations for most of variables. This could further limit our research findings’ application in other study areas. We believe that with larger sample size, models would show a larger R2 and significant associations for more independent variables. Models with larger sample size could be generalized to a larger study area. With these limitations, our study findings are still valid since our sensitivity analysis showed stability of these results (S Table 1-2). Future studies can consider other survey methods to improve the representativeness of the sample.” 

[Comment 3] When measuring actual pollution exposure, PM2.5 is measured by the average mean value of exposure for the two survey days (the two days include one weekday and one

weekend day) in this study. Would it be more accurate to consider weekdays and weekends separately? Given that weekdays make up five out of seven days of the week, could a weighting system be implemented to address this issue?

(Response 3) Thanks for your suggestion. We used linear regression to learn the potential difference of time-weighted exposure and average mean value of exposure on our results. The result showed that there is a high correlation between the two methods (coefficient: 0.923, R2: 0.962). Therefore, the result showed that using time-weighted method and average mean value would not generate significant difference in our result. In other words, our results demonstrated that using different exposure estimated methods (i.e., time-weighted exposure or average mean value of exposure) would not significantly influence our conclusions. 

[Comment 4] Since apply RB and MB approaches relying on different regression models to assess people's multiple environmental exposures, may produce divergent findings regarding the associations between people's perceived air pollution, their environmental exposures, and socio-demographic attributes. In the future, which model should be selected, and how can policy recommendations be effectively formulated?

(Response 4) Thanks for the comments. We have added more discussion on the comparison of RB and MB methods and policy recommendations in the revised manuscript. 

(Page 33-34, Line 571-592)

“These results have important implications. First, our study found that the association between environmental factors and perception of air pollution could be different with RB and MB measurements used. Therefore, studies on mobility-dependent exposures need to recognize UGCoP by considering people’s daily mobility. Environmental health research with RB measurements neglects spatial and temporal variations of mobility-dependent exposures, which may lead to inaccurate results. Second, ignoring mobility and failing to address UGCoP might mislead policymakers when researchers explore the association between mobility-dependent environmental factors and perception of air pollution. For example, our study found that people with an older age tend to underestimate air pollution only in the MB model, not in the RB model. Therefore, public air pollution education (such as the nature and magnitude of air pollution problems, emission sources, and related health risks) should be held, especially towards older people who tend to underestimate air pollution. This could increase people’s basic knowledge about air pollution, rectify the misunderstanding of air pollution, and enhance comprehension and cooperation among the public, environmental departments, and academic communities [69]. Without using MB method, policies might lose focus and have low efficiency. Using the MB method, our study also found that transportation land-use density and open space density relate to people’s perception of air pollution. Therefore, policymaking should consider the psychological, cognitive, and mental health effects of landscapes when planning our environment. Besides, people’s perceptions of the environment are quite momentary and influenced by specific microclimates. Real-time and high-resolution spatial information on air pollution could be provided to guide people’s daily activities and travel to reduce their frequency and duration at high air-pollution microenvironments.”

Reference:

[69] Liu X, Zhu H, Hu Y, Feng S, Chu Y, Wu Y, et al. Public’s Health Risk Awareness on Urban Air Pollution in Chinese Megacities: The Cases of Shanghai, Wuhan and Nanchang. International Journal of Environmental Research and Public Health. 2016;13(9):845. PubMed PMID: doi:10.3390/ijerph13090845.

---

## [Decision Letter · Decision Letter 1]

13 Jun 2023

PONE-D-23-04918R1Assessment of Air Pollution and Air Quality Perception Mismatch Using Mobility-based Real-time ExposurePLOS ONE

Dear Dr. Kwan,

Thank you for submitting your manuscript to PLOS ONE. After careful consideration, we feel that it has merit but does not fully meet PLOS ONE’s publication criteria as it currently stands. Therefore, we invite you to submit a revised version of the manuscript that addresses the points raised during the review process.

We look forward to receiving your revised manuscript.

Kind regards,

Tai Ming Wut

Academic Editor

PLOS ONE

Journal Requirements:

Reviewers' comments:

Reviewer's Responses to Questions

**Comments to the Author**

1. If the authors have adequately addressed your comments raised in a previous round of review and you feel that this manuscript is now acceptable for publication, you may indicate that here to bypass the “Comments to the Author” section, enter your conflict of interest statement in the “Confidential to Editor” section, and submit your "Accept" recommendation.

Reviewer #3: (No Response)

Reviewer #4: (No Response)

2. Is the manuscript technically sound, and do the data support the conclusions?

Reviewer #3: (No Response)

Reviewer #4: Yes

3. Has the statistical analysis been performed appropriately and rigorously? 

Reviewer #3: (No Response)

Reviewer #4: Yes

4. Have the authors made all data underlying the findings in their manuscript fully available?

Reviewer #3: (No Response)

Reviewer #4: Yes

5. Is the manuscript presented in an intelligible fashion and written in standard English?

Reviewer #3: (No Response)

Reviewer #4: Yes

6. Review Comments to the Author

Reviewer #3: (No Response)

Reviewer #4: I did not review the initial manuscript, but see the auhthors have been responsive to comments. I will only give some comments to improve clarity of the paper, which is well written and done.

1. the number of tables is large and some are not that informative. Table 6 could be included in tables 5. Tables 3 and 4 could be combined and shortened for the main text table and rest to a supplement. Table 2 can be summarized in the text.

2. Add justification for using linear regression with a ordinal variable (perceived air q) as dependent. It is acceptable but associated with assumptions.

3. Some tables and figures need a bit more legend e.g. just add meaning of abbreviations MB=mobility-based, easier for the reader. In figure 1: Land use of the study area (Hong Kong)

4. In the abstract explain mismatch (figure 4 nicely shows that), add the dichotomization in the abstract. The choice of 15 for PM2.5 needs abit more explanation (natural breaks??). Figure 4 illustrates it is quite reasoanble.

7. PLOS authors have the option to publish the peer review history of their article (what does this mean?). If published, this will include your full peer review and any attached files.

Reviewer #3: No

Reviewer #4: No

---

## [Author Response · Author response to Decision Letter 1]

5 Jul 2023

Manuscript ID: PONE-D-23-04918 [EMID:0f802a08040d887a]

“Assessment of Air Pollution and Air Quality Perception Mismatch Using Mobility-based Real-time Exposure”

Response to Reviewers’ comments

Dear Editor,

We appreciate your decision letter and the helpful feedback from the reviewers on our manuscript. We have carefully revised our paper to address their comments and have made corrections to address their concerns. We have submitted a version of our revised manuscript with “Tacked Changes” and an unmarked version without changes. 

We are particularly grateful to you and the reviewers for your efforts in handling and reviewing our paper. The thoughtful comments provided by the reviewers have greatly assisted us in improving the paper. We look forward to hearing from you regarding our revised manuscript.

Best regards,

The Authors

Journal Requirements:

[Comment 1] Please review your reference list to ensure that it is complete and correct. If you have cited papers that have been retracted, please include the rationale for doing so in the manuscript text, or remove these references and replace them with relevant current references. Any changes to the reference list should be mentioned in the rebuttal letter that accompanies your revised manuscript. If you need to cite a retracted article, indicate the article’s retracted status in the References list and also include a citation and full reference for the retraction notice.

(Response 1) The authors have carefully checked the format of references and changed the first reference to the following status. We found no retracted papers in our references. 

(Page 37, Line 665-667)

“1. Yap J, Ng Y, Yeo KK, Sahlen A, Lam CSP, Lee V, et al. Particulate air pollution on cardiovascular mortality in the tropics: impact on the elderly. Environmental Health. 2019;18. doi: 10.1186/s12940-019-0476-4.”

Reviewer(s)' Comments to Author:

Reviewer #4: I did not review the initial manuscript, but see the authors have been responsive to comments. I will only give some comments to improve clarity of the paper, which is well written and done.

Specific comments:

[Comment 1] The number of tables is large, and some are not that informative. Table 6 could be included in tables 5. Tables 3 and 4 could be combined and shortened for the main text table and rest to a supplement. Table 2 can be summarized in the text.

(Response 1) Thanks for your suggestions. The authors have deleted Table 6 and combined it with Table 5 and the new table’s name is “Table 3”. We have also deleted Table 4 and combined it with Table 3, named as “Table 2” and moved some independent variables to the supplementary file, named as “S1 Table”. Additionally, we have updated the table numbers in the main text and supplementary file. 

(Page 20-21, Line 365-366)

Table 3. Description of residence-based (RB) and mobility-based (MB) environmental factors and Mann-Whitney U test significance.

Environmental exposures ring buffer Mean Std. Min Max Sig.

Transportation land-use density

 RB 100m-150m 3.10E-03 1.20E-03 2.30E-04 5.70E-03 <0.001

 MB 350m-400m 2.70E-01 6.30E-02 7.30E-02 4.50E-01 

Green space density

 RB 100m-150m 1.30E-04 3.80E-04 0.00E+00 2.90E-03 <0.001

 MB 350m-400m 8.30E-02 6.70E-02 1.50E-03 3.50E-01 

Open space density

 RB 100m-150m 6.30E-04 8.50E-04 0.00E+00 4.40E-03 <0.001

 MB 350m-400m 1.00E-01 6.10E-02 5.60E-03 3.40E-01 

Facility density

 RB 100m-150m 1.70E-04 1.50E-04 0.00E+00 7.60E-04 0.021

 MB 350m-400m 1.30E-04 7.40E-05 1.20E-05 3.10E-04 

Population density RB - 1.30E+05 6.60E+04 6.00E+03 3.50E+05 -

(Page 19-20, Line 348)

Table 2. Descriptive statistics of participants’ characteristics.

Personal attributes Code Percent

 SSP

 TSW

 Sample Census Sample Census 

Gender 

Male 0 45.2% 43.8% 46.7% 46.7% 46.3%

Female

 1 54.8% 56.2% 53.3%

 53.3% 53.7%

Age 

18-24 - 18.6% 15.2% 14.3% 21.9% 15.7%

25-44 - 48.1% 47.6% 42.1% 48.6% 38.7%

45-64 - 33.3% 37.1% 43.6%

 29.5% 45.6%

Household Income (HKD) 

< 9999 1 9.0% 11.4% 23.1% 6.7% 19.7%

10000 - 19999 2 29.0% 35.2% 31.7% 22.9% 25.8%

20000 - 39999 3 37.1% 31.4% 27.3%

 42.9% 34.4%

>40000 4 24.8% 21.9% 17.8%

 27.6% 20.0%

Employment status 

Employed 0 75.7% 78.1% 74.5%

 73.3% 78.6%

Student 1 11.0% 8.6% 14.3%

 13.3% 6.7%

Unemployed 1 13.3% 13.3% 11.2%

 13.3% 14.8%

Education 

Primary and below 1 2.4% 1.9% 2.9%

Secondary 2 33.8% 35.2% 32.4%

Post-secondary: Diploma/ Certificate 3 8.6% 11.4% 5.7%

Post-secondary: Sub-degree course 4 3.8% 4.8% 2.9%

Post-secondary: Degree course 5 40.0% 36.2% 43.8%

Master and above 6 11.4% 10.5% 12.4%

Marital Status 

Never Married 0 52.9% 49.5% 56.2%

Divorced 1 8.1% 7.6% 8.6%

Married 1 36.7% 40.0% 33.3%

Widowed 1 2.4% 2.9% 1.9%

Separated 1 0.0% 

(Supplementary file)

S1 Table. Descriptive statistics of participants’ characteristics.

Personal attributes Code Percent SSP

 TSW

House ownership 

Rent 0 59.0% 62.9% 55.2%

Own without mortgage 1 21.0% 20.0% 21.9%

Own with mortgage 1 19.0% 15.2% 22.9%

Living space (ft2) 

<100 1 3.3% 6.7% 

100-300 2 28.1% 37.1% 19.0%

301-500 3 44.3% 45.7% 42.9%

501-800 4 21.4% 9.5% 33.3%

801-1100 5 1.4% 1.0% 1.9%

>1100 6 1.4% 2.9%

Family member 

1-2 - 29.5% 42.9% 16.2%

3-4 - 59.0% 50.5% 67.6%

5-6 - 11.4% 6.7% 16.2%

Respiratory symptoms 

0-6 - 46.7% 38.1% 55.2%

7-12 - 48.6% 59.0% 38.1%

13-18 - 2.9% 0.0% 5.7%

Physical exercise days in one week (Physical exercise) 

≤3 - 82.9% 87.6% 78.1%

4-7 - 17.1% 12.4% 21.9%

[Comment 2] Add justification for using linear regression with an ordinal variable (perceived air q) as dependent. It is acceptable but associated with assumptions.

(Response 2) Thanks for the comment. The authors added the assumption and showed the results in the main text. 

(Page 15, Line 286-288)

“Since the dependent variable is normally distributed, as shown in Table 1 and verified by the Shapiro-Wilks test,”

[Comment 3] Some tables and figures need a bit more legend e.g. just add meaning of abbreviations MB=mobility-based, easier for the reader. In figure 1: Land use of the study area (Hong Kong)

(Response 3) Thanks for your suggestions. We added more words to provide clearer descriptions in the titles of figure and tables. 

(Page 9, Line 184)

“Fig 1. Land use of the study area (Hong Kong)”

(Page 20, Line 365-366)

“Table 3. Description of residence-based (RB) and mobility-based (MB) environmental factors and Mann-Whitney U test significance.”

(Page 23, Line 403-404)

“Table 5. Regression results of perceived air pollution (Model 1: residence-based method (RB) and Model 2: mobility-based (MB) method).”

(Page 26, Line 449-450)

“Table 6. Logistic regression of perception mismatch with normal perception (high-high) as the reference group (Model 3: residence-based method (RB) and Model 4: mobility-based (MB) method).”

[Comment 4] In the abstract explain mismatch (figure 4 nicely shows that), add the dichotomization in the abstract. The choice of 15 for PM2.5 needs a bit more explanation (natural breaks??). Figure 4 illustrates it is quite reasonable.

(Response 4) Thanks for your comments. We added the definition of mismatch in the abstract. The authors replaced “natural breaks” with “K-means clustering” in main text to make this method more easily understood by readers. 

(Page 2, Line 24-27)

“Mismatch is defined as the perception of poor air quality when the air pollution level is smaller than 15.536μg/m3 and the perception of good air quality when the air pollution level is larger than 15.608μg/m3.”

(Page 16, Line 294-295)

“Actual air quality is divided into two groups (low: 3.208μg/m³ - 15.536μg/m³, high: 15.608μg/m³-38.834μg/m³) based on using K-means clustering.”

---

## [Decision Letter · Decision Letter 2]

14 Sep 2023

PONE-D-23-04918R2Assessment of Air Pollution and Air Quality Perception Mismatch Using Mobility-based Real-time ExposurePLOS ONE

Dear Dr. Kwan,

Thank you for submitting your manuscript to PLOS ONE. After careful consideration, we feel that it has merit but does not fully meet PLOS ONE’s publication criteria as it currently stands. Therefore, we invite you to submit a revised version of the manuscript that addresses the points raised during the review process.

We look forward to receiving your revised manuscript.

Kind regards,

Tai Ming Wut

Academic Editor

PLOS ONE

Journal Requirements:

Reviewers' comments:

Reviewer's Responses to Questions

**Comments to the Author**

1. If the authors have adequately addressed your comments raised in a previous round of review and you feel that this manuscript is now acceptable for publication, you may indicate that here to bypass the “Comments to the Author” section, enter your conflict of interest statement in the “Confidential to Editor” section, and submit your "Accept" recommendation.

Reviewer #5: (No Response)

Reviewer #6: (No Response)

2. Is the manuscript technically sound, and do the data support the conclusions?

Reviewer #5: Yes

Reviewer #6: Yes

3. Has the statistical analysis been performed appropriately and rigorously? 

Reviewer #5: Yes

Reviewer #6: Yes

4. Have the authors made all data underlying the findings in their manuscript fully available?

Reviewer #5: No

Reviewer #6: No

5. Is the manuscript presented in an intelligible fashion and written in standard English?

Reviewer #5: Yes

Reviewer #6: Yes

6. Review Comments to the Author

Reviewer #5: Review of PONE-D-23-04918R2 "Assessment of Air Pollution and Air Quality Perception Mismatch Using Mobility-based Real-time Exposure" by Wanying Song et al

The authors present a cohort study on perceived air pollution and observed air pollution in Hong Kong, including different social groups. They use residential and a mobility-based geographic context approach to compare pollution exposure, several statistical models are used to analyse potential effects. The manuscript is clearly structured, well written and should be understandable for non-experts of the topic. The presented material fits in the scope of the journal and can be of interest for the readers.

I did not review the initial version, but the authors responded properly to previous reviewer's comments, and the manuscript is already in a good shape. I have a few items that require clarification (I opted for major revision as there is no moderate). I hope the authors can incorporate those into the manuscript before it can be published. I hope you will find my suggestions useful.

Your data was achieved in summer 2021, which falls in the COVID-19 pandemic. To what extent does that influence the mobility pattern of the participants? Do weekdays reflect the normal working pattern? Is there sufficient difference in (mobile) activities between weekdays and weekends? Similar, how did air pollution in that area change compared to non-pandemic times? Were industrial activities and thus pollution reduced at the time of study?

Minor question, why did you create 1415x1415m grids for the facilities, while e.g. land use was on 10x10m? 1415 seems somewhat unusual as cell size...

Figure 1: The two areas are hard to identify in the left side of the figure. Perhaps you could increase the line thickness of the boundaries, or place TSW and SSP close to the areas to make them easier to find.

line 609: RH occurs first here, I assume you mean relative humidity? If so, perhaps just use that.

Reviewer #6: • Consider this statement from abstract: “A mismatch happens when the perceived air quality is poor but the air pollution level is lower than 15.536μg/m3 and when the perceived air quality is good but the air pollution level is higher than 15.608μg/m3.” Are the authors saying that some how pollution levels below 15.5 μg/m3 is objectively low and pollution levels above 15.6 μg/m3 are objectively high? How is this cutoff of 15.5 μg/m3 arrived at? From methods, it seems k-means clustering was used hence mentioning this as a justification for the cutoff may be warranted in the abstract.

• Rewrite this sentence in abstract as it is not clear what the authors mean: “People with higher exposures to MB transportation land use density and green space tend to have negative assessments of air pollution.”

• It is not clear what the authors are trying to say in lines 64-72. For example, in their own words, “People with lower levels of education and income were found to be likely to perceive poor air quality” and “Well-educated people were likely to be more informed about the adverse health effects of air pollution, which may lead to higher levels of perceived risk”. To me it reads like both well educated and low educated people are able to perceive low air quality levels. But perhaps authors are saying that “low educated people can perceive bad air quality but do not have sufficient information to perceive the associated risk”? Overall, I think this paragraph needs to be more crisp.

• Several studies looked at the issues in lines 109-112 but authors do not cite any studies. Example studies include Setton et al. 2011 and Gurram et al. 2015.

7. PLOS authors have the option to publish the peer review history of their article (what does this mean?). If published, this will include your full peer review and any attached files.

Reviewer #5: No

Reviewer #6: No

---

## [Author Response · Author response to Decision Letter 2]

3 Oct 2023

PONE-D-23-04918R2

Assessment of Air Pollution and Air Quality Perception Mismatch Using Mobility-based Real-time Exposure

PLOS ONE

Journal Requirements:

(Response) We have revised the reference list accordingly. 

Reviewers' comments:

Comments to the Author

1. If the authors have adequately addressed your comments raised in a previous round of review and you feel that this manuscript is now acceptable for publication, you may indicate that here to bypass the “Comments to the Author” section, enter your conflict of interest statement in the “Confidential to Editor” section, and submit your "Accept" recommendation.

Reviewer #5: (No Response)

Reviewer #6: (No Response)

2. Is the manuscript technically sound, and do the data support the conclusions?

Reviewer #5: Yes

Reviewer #6: Yes

3. Has the statistical analysis been performed appropriately and rigorously?

Reviewer #5: Yes

Reviewer #6: Yes

4. Have the authors made all data underlying the findings in their manuscript fully available?

Reviewer #5: No

Reviewer #6: No

(Response) Our data cannot be shared publicly because they contain potentially identifying information that cannot be fully de-identified. Note that the dataset has participants’ GPS tracks and activity locations (including home and workplace locations), which can be used to discover the identity of individual participants using a process called spatial reverse engineering (e.g., by linking high-resolution geospatial data with other data such as census or survey data to discover the identity of specific individuals). The Survey and Behavioral Research Ethics (SBRE) Committee of the Chinese University of Hong Kong has imposed this restriction. Please contact the SBRE at fssc02@cuhk.edu.hk for inquiries about data availability or access.

5. Is the manuscript presented in an intelligible fashion and written in standard English?

Reviewer #5: Yes

Reviewer #6: Yes

6. Review Comments to the Author

Reviewer #5: Review of PONE-D-23-04918R2 "Assessment of Air Pollution and Air Quality Perception Mismatch Using Mobility-based Real-time Exposure" by Wanying Song et al

The authors present a cohort study on perceived air pollution and observed air pollution in Hong Kong, including different social groups. They use residential and a mobility-based geographic context approach to compare pollution exposure, several statistical models are used to analyse potential effects. The manuscript is clearly structured, well written and should be understandable for non-experts of the topic. The presented material fits in the scope of the journal and can be of interest for the readers.

I did not review the initial version, but the authors responded properly to previous reviewer's comments, and the manuscript is already in a good shape. I have a few items that require clarification (I opted for major revision as there is no moderate). I hope the authors can incorporate those into the manuscript before it can be published. I hope you will find my suggestions useful.

[Comment 1] Your data was achieved in summer 2021, which falls in the COVID-19 pandemic. To what extent does that influence the mobility pattern of the participants? Do weekdays reflect the normal working pattern? Is there sufficient difference in (mobile) activities between weekdays and weekends? Similar, how did air pollution in that area change compared to non-pandemic times? Were industrial activities and thus pollution reduced at the time of study?

(Response 1) Thank you for your suggestion. We added text on the COVID-19 pandemic and the corresponding air pollution concentration changes during our survey time in the revised manuscript to address your concerns. 

(Pages 10-11, Lines 203-213)

“During our survey, which was conducted in low-risk periods of COVID-19 in Hong Kong, there were no significant mobility restrictions compared to previous waves. When compared to the baseline day (the median value for the five-week period of January 3 to February 6, 2020), COVID-19 community mobility reports showed a 36.5% increase in visits to grocery and pharmacy stores, a 6.4% increase in visits to residential areas, a 9.3% decrease in visits to retail and recreation venues, a 7.3% decrease in visits to transit stations, an 18.1% decrease in visits to parks, and a 5.8% decrease in visits to workplaces on average [49]. In 2020, industrial activities decreased, air pollution concentrations decreased, and air quality improved due to COVID-19 [50]. However, as the epidemic stabilized in 2021, domestic economic activities and electricity demand rebounded, resulting in similar average annual levels of PM2.5 (recorded by stationary monitoring stations) and increased levels of PM2.5 (recorded by roadside stations) compared to 2020 [50, 51].”

References: 

49. COVID-10 Community Mobility Reports. Available at https://www.google.com/covid19/mobility/. Visited 29th Sep 2023. [Internet]. 2021.

50. HKSARG. Hong Kong greenhouse gas emission inventory for 2021. Available at https://www.info.gov.hk/gia/general/202307/04/P2023070300212.htm. Visited 29th Sep 2023. 2021.

51. HKSARG. 2013 to 2022 Annual Average Concentrations of Major Air Pollutants in Hong Kong. Available at https://gia.info.gov.hk/general/202302/09/P2023020900560_412646_1_1675938001355.pdf. Visited 29th Sep 2023. 2023.

[Comment 2] Figure 1: The two areas are hard to identify in the left side of the figure. Perhaps you could increase the line thickness of the boundaries, or place TSW and SSP close to the areas to make them easier to find.

(Response 2) Thanks for your suggestion. We increased the line thickness of the boundaries.

(Page 9, Line 184)

[Comment 2] Minor question, why did you create 1415x1415m grids for the facilities, while e.g. land use was on 10x10m? 1415 seems somewhat unusual as cell size...

(Response 2) Thanks for your comment. The grid size was determined to reduce the processing power required when obtaining facility locations. The Nearby API can generate facilities within a 1km radius of a given location. The diagonal of a grid with a 1km width is approximately 1415 meters. Creating grids with a width of 1415 meters minimizes the spatial overlap of facilities. That is why we chose this grid size. The grid size was only used for obtaining facility information. We revised our expression to make it clearer in the revised manuscript. 

(Page 15, Lines 274-276)

“We covered the spatial extent of HK with 1415m×1415m grids (note that the diagonal of a grid with a 1km width is approximately 1415 meters). We then obtained the locations of all facilities in these grids and then measured RB and MB exposures.”

[Comment 3] line 609: RH occurs first here, I assume you mean relative humidity? If so, perhaps just use that.

(Response 3) Thanks for your comment. We added the abbreviation of relative humidity “RH” at the first place this word appears. 

(Page 7, Line 145)

“relative humidity (RH)”

 

Reviewer #6: 

[Comment 1] Consider this statement from abstract: “A mismatch happens when the perceived air quality is poor but the air pollution level is lower than 15.536μg/m3 and when the perceived air quality is good but the air pollution level is higher than 15.608μg/m3.” Are the authors saying that some how pollution levels below 15.5 μg/m3 is objectively low and pollution levels above 15.6 μg/m3 are objectively high? How is this cutoff of 15.5 μg/m3 arrived at? From methods, it seems k-means clustering was used hence mentioning this as a justification for the cutoff may be warranted in the abstract.

(Response 1) Thanks for your comment. We mentioned using K-means clustering to classify PM2.5 value range into 2 groups. 

(Page 2, Line 26-27)

“using K-means clustering to divide PM2.5 value range into two groups”

[Comment 2] Rewrite this sentence in abstract as it is not clear what the authors mean: “People with higher exposures to MB transportation land use density and green space tend to have negative assessments of air pollution.”

(Response 2) Thanks for your comment. We revised this sentence. 

(Page 2, Lines 35-36)

“People who are surrounded by higher MB transportation land-use density and green space density tend to perceive air quality poor”

[Comment 3] It is not clear what the authors are trying to say in lines 64-72. For example, in their own words, “People with lower levels of education and income were found to be likely to perceive poor air quality” and “Well-educated people were likely to be more informed about the adverse health effects of air pollution, which may lead to higher levels of perceived risk”. To me it reads like both well educated and low educated people are able to perceive low air quality levels. But perhaps authors are saying that “low educated people can perceive bad air quality but do not have sufficient information to perceive the associated risk”? Overall, I think this paragraph needs to be more crisp.

(Response 3) Thanks for your comments. After carefully checking, we deleted this sentence “People with lower levels of education and income were found to be likely to perceive poor air quality” to make this paragraph concise. 

[Comment 4] Several studies looked at the issues in lines 109-112 but authors do not cite any studies. Example studies include Setton et al. 2011 and Gurram et al. 2015.

(Response 4) Thanks for your comment. We added the two references in our revised manuscript. 

(Pages 5-6, Lines 108-114)

“However, using people’s residential location as the geographic context may not reveal how much air pollution they are actually exposed to because people travel to various locations and are also exposed to these non-residential contexts [25, 26]. Using only the residential location thus cannot capture how people interact with and interpret the air quality in different geographic or activity contexts [9]. People’s awareness of good or poor air quality may be influenced by environmental factors in the places they visited and at locations other than their residential neighborhoods.”

References:

9. Bickerstaff K, Walker G. Public understandings of air pollution: the ‘localisation’ of environmental risk. Global Environmental Change. 2001;11(2):133-145. doi: https://doi.org/10.1016/S0959-3780(00)00063-7.

25. Gurram S, Stuart AL, Pinjari AR. Impacts of travel activity and urbanicity on exposures to ambient oxides of nitrogen and on exposure disparities. Air Quality, Atmosphere & Health. 2015;8(1):97-114. doi: 10.1007/s11869-014-0275-6.

26. Setton E, Marshall JD, Brauer M, Lundquist KR, Hystad P, Keller P, et al. The impact of daily mobility on exposure to traffic-related air pollution and health effect estimates. Journal of Exposure Science & Environmental Epidemiology. 2011;21(1):42-48. doi: 10.1038/jes.2010.14.

7. PLOS authors have the option to publish the peer review history of their article (what does this mean?). If published, this will include your full peer review and any attached files.

Do you want your identity to be public for this peer review? For information about this choice, including consent withdrawal, please see our Privacy Policy.

Reviewer #5: No

Reviewer #6: No

In addition to the above revisions, we shortened the digits in table 4.

(Page 22, Lines 385)

Table 4. ANOVA analysis for four factors.

 Gender Age Income Employment status

Anova sig 0.534 0.204 0.079 0.008

---

## [Decision Letter · Decision Letter 3]

5 Nov 2023

Assessment of Air Pollution and Air Quality Perception Mismatch Using Mobility-based Real-time Exposure

PONE-D-23-04918R3

Dear Mei-Po Kwan,

We’re pleased to inform you that your manuscript has been judged scientifically suitable for publication and will be formally accepted for publication once it meets all outstanding technical requirements.

Kind regards,

Tai Ming Wut

Academic Editor

PLOS ONE

Additional Editor Comments (optional):

Reviewers' comments:

Reviewer's Responses to Questions

**Comments to the Author**

1. If the authors have adequately addressed your comments raised in a previous round of review and you feel that this manuscript is now acceptable for publication, you may indicate that here to bypass the “Comments to the Author” section, enter your conflict of interest statement in the “Confidential to Editor” section, and submit your "Accept" recommendation.

Reviewer #6: All comments have been addressed

2. Is the manuscript technically sound, and do the data support the conclusions?

Reviewer #6: Yes

3. Has the statistical analysis been performed appropriately and rigorously? 

Reviewer #6: Yes

4. Have the authors made all data underlying the findings in their manuscript fully available?

Reviewer #6: Yes

5. Is the manuscript presented in an intelligible fashion and written in standard English?

Reviewer #6: Yes

6. Review Comments to the Author

Reviewer #6: (No Response)

7. PLOS authors have the option to publish the peer review history of their article (what does this mean?). If published, this will include your full peer review and any attached files.

Reviewer #6: No

---

## [Editor Report · Acceptance letter]

10 Nov 2023

PONE-D-23-04918R3 

Assessment of Air Pollution and Air Quality Perception Mismatch Using Mobility-based Real-time Exposure 

Dear Dr. Kwan:

I'm pleased to inform you that your manuscript has been deemed suitable for publication in PLOS ONE. Congratulations! Your manuscript is now with our production department. 

Kind regards, 

on behalf of

Dr. Tai Ming Wut 

Academic Editor

PLOS ONE